



# How Does Downward Planetary Wave Coupling Affect Polar Stratospheric Ozone in the Arctic Winter Stratosphere?

Sandro W. Lubis[1], Vered Silverman[2], Katja Matthes[1, 3], Nili Harnik[2, 4], Nour-Eddine Omrani[5], and Sebastian Wahl[1]

[1]GEOMAR Helmholtz Centre for Ocean Research Kiel, Germany
[2]Department of Geophysical, Atmospheric and Planetary Sciences, Tel Aviv University, Israel
[3]Christian-Albrechts Universität zu Kiel, Germany
[4]Department of Meteorology, University of Stockholm, Sweden
[5]Geophysical Institute, University of Bergen and Bjerknes Centre for Climate Research, Norway

*Correspondence to:* S. W. Lubis (slubis@geomar.de)

**Abstract.**

It is well established that variable wintertime planetary wave forcing in the stratosphere controls the variability of Arctic stratospheric ozone through changes in the strength of the polar vortex and the residual circulation. While previous studies focused on the variations in upward wave flux entering the lower stratosphere, here the impact of downward planetary wave coupling (DWC) on ozone is investigated for the first time. Utilizing the MERRA2 reanalysis and a fully coupled chemistry-climate simulation with NCAR's Community Earth System Model (CESM1[WACCM]), we find two DWC effects on ozone: (1) the direct effect in which the residual circulation is modified and prevents the typical increase of ozone due to upward planetary wave events in winter, and (2) the indirect effect in which the modification of polar temperature during winter affects the amount of ozone destruction in spring.

Winter seasons dominated by DWC events (i.e., reflective winters) are characterized by lower Arctic ozone concentration, while seasons dominated by increased upward wave events (i.e., absorptive winters) are characterized by relatively higher ozone concentration. This behavior is consistent with the cumulative effects of downward and upward planetary wave events on polar stratospheric ozone via the residual circulation and the polar temperature in winter. The results establish a new perspective on dynamical processes controlling stratospheric ozone variability in the Arctic.

## 1 Introduction

The dynamical linkage between the stratosphere and troposphere is dominated by planetary waves, which are generated in the troposphere by orographic and/or nonorographic forcing (e.g., Kuroda and Kodera 1999; Christiansen 2001; Plumb and Semeniuk 2003; Polvani and Waugh 2004). These waves propagate upward into the stratosphere where they either dissipate (often manifested as a sudden stratospheric warming -SSW) and initiate downward zonal-mean coupling (e.g., Baldwin and Dunkerton 2001; Limpasuvan et al. 2004); or are reflected downward toward the troposphere, which results in downward wave coupling (DWC) (Perlwitz and Harnik 2003; Shaw et al. 2010; Lubis et al. 2016a). The DWC occurs when upward pulses of





wave activity decelerate the flow in the upper stratosphere, forming a downward-reflecting surface that redirects waves back to the troposphere (see Harnik and Lindzen 2001, for theoretical considerations). The occurrence of DWC is tied to the so-called bounded wave geometry of the stratospheric basic state, which is characterized by a well-defined high-latitude meridional waveguide in the lower stratosphere and a vertical reflecting surface in the upper stratosphere (e.g., Shaw et al. 2010; Lubis et al. 2016a).

The vertical coupling of planetary-scale waves between the stratosphere and troposphere can be directly examined via the meridional eddy heat flux, as it represents the vertical group velocity of the planetary waves (e.g., Shaw and Perlwitz 2013; Lubis et al. 2016a). Using extreme total (climatology plus anomaly) negative wave-1 heat flux values, the life cycle of DWC and its subsequent impact on the circulation in the Northern Hemisphere (NH) have been studied extensively (e.g., Shaw and Perlwitz 2013, 2014; Dunn-Sigouin and Shaw 2015; Lubis et al. 2016a). Shaw and Perlwitz (2014) showed that the occurrence of DWC events coincide with a transient reversal of the residual circulation in the stratosphere. This result is consistent with Eliassen-Palm (EP) flux divergence in the stratosphere, induced by transient downward wave propagation during the event (Dunn-Sigouin and Shaw, 2015; Lubis et al., 2016a). Since the variations in the stratospheric EP flux divergence results in changes in the residual circulation (Plumb, 2002), it is therefore expected that DWC may influence the Arctic ozone levels by changing the dynamical transport of ozone to the pole. This hypothesis will be tested in this study.

It is well established that planetary waves play an important role in shaping the ozone hole through their impact on the polar vortex and on the residual circulation (e.g., Solomon 1999; Fusco and Salby 1999; Randel et al. 2002). Randel et al. (2002) showed that variations in planetary wave forcing in the lower stratosphere during winter-spring exhibit a strong correlation with column ozone. The mechanism for this is that increases (decreased) wave dissipation in the stratosphere leads to strengthening (weakening) of residual circulation, which in turn increases (decreases) transport of ozone-rich air to the polar lower stratosphere. On the other hand, strong (weak) winter planetary wave forcing causes a warmer (cooler) Arctic lower stratosphere in early spring (Newman et al., 2001), resulting in smaller (larger) chemical ozone losses in spring. Manney et al. (2011) reveal that the unprecedented large Arctic ozone loss in 2011 is highly correlated with extremely cold lower-stratospheric temperatures in early spring. These extremely low temperatures are attributed to the unusually weak midwinter planetary wave forcing in the stratosphere (Hurwitz et al., 2011), as expected from a close relationship between polar spring temperatures and eddy heat flux in mid to late winter (Newman et al., 2001). Both of these dynamical ozone supply and chemical ozone losses are equally important for the variability of column ozone in the Arctic winter stratosphere (Tegtmeier et al., 2008).

Weaker planetary wave driving in the stratosphere, which affects ozone through both dynamical and chemical processes, could arise from an enhanced number of extreme negative wave-1 heat flux events (i.e., DWC events), or from anomalously low positive heat flux values. Many studies have shown that an increased number of major SSW events that are associated with enhanced upward wave propagation into the stratosphere have led to significant increases of total column ozone and polar temperature in the winter, and subsequently less springtime ozone destruction (e.g., Rose and Brasseur 1985; Randel 1993; Liu et al. 2011; Hocke et al. 2015). However, the impact of downward wave propagation associated with planetary wave reflection on the ozone in the Arctic winter stratosphere has never been explored. Understanding the impact of DWC on





residual circulation and polar temperatures will help to improve our knowledge of the links between stratospheric dynamics and ozone variability.

The goal of this study is to investigate the impact of DWC on polar stratospheric ozone in the NH using both Modern-Era Retrospective analysis for Research and Applications, Version 2 (MERRA2, Bosilovich et al. 2015) and the NCAR's Commu-

nity Earth System Model, Version 1.0.2 (CESM1 [WACCM]) model simulation. The new MERRA2 reanalysis improves on and augments the original MERRA reanalysis (Bosilovich et al., 2015). Therefore, it has a better representation of stratospheric ozone in high latitudes compared to the original MERRA renalaysis (see Appendix A1, Figs. A1 and A2). On the other hand, CESM1(WACCM) model is able to capture the main features of DWC in the NH winter (Lubis et al., 2016a), therefore it can be used to study the link between DWC and polar stratospheric ozone. Here, we focus on two kinds of DWC effects on ozone:

(1) the direct effect, which is analyzed over the whole life cycle of the individual DWC event, and (2) indirect effect, which is analyzed through the seasonal impact of DWC on polar temperature.

The paper is structured as follows: Section 2 describes the data and methods. This includes the description of data, model simulation, and methods. In Section 3, the direct effects of DWC on the mean residual transport, temperature and ozone concentrations are analyzed, based on MERRA2 reanalysis product and a 100-yr transient simulation from the fully coupled

chemistry-climate model CESM1(WACCM). Section 4 discusses the seasonal impacts of DWC events on the Arctic column ozone during seasons that are dominated by enhanced DWC events (reflective winters) and by enhanced upward wave events (absorptive winters). Finally, the results are summarized and discussed in section 5.

## 2    Data and Methods

### 2.1    MERRA2 Ozone

The new MERRA2 daily ozone product from 1980 to 2013 (Bosilovich et al., 2015) was used to investigate the impact of DWC on ozone. The data was stored on 42 pressure levels from the surface up to 0.1 hPa. MERRA2 assimilates ozone from the Solar Backscattered Ultra Violet (SBUV) radiometers from October 1978 to October 2004, and thereafter from the Ozone Monitoring Instrument (OMI) and AURA MLS (Microwave Limb Sounder) instrument (Bosilovich et al., 2015). These ozone datasets are assimilated with the Goddard Earth Observing System Model, Version 5 (GEOS-5) by using odd-oxygen mixing

ratio ($qO_x$) as its prognostic variable (Bosilovich et al., 2015). This includes an odd-oxygen family transport model providing the ozone concentration necessary for solar absorption. The vertically integrated ozone tendency is given as (Bosilovich et al., 2015) :

$$\frac{\partial \overline{qO_x}}{\partial t} = \left[ -\nabla \bullet (\overline{\boldsymbol{v}\,qO_x}) \right]_{DYN} + \left[ \frac{\partial \overline{qO_x}}{\partial t} \right]_{PHY} + \left[ \frac{\partial \overline{qO_x}}{\partial t} \right]_{ANA} \qquad (1)$$

The dynamical contribution to the total ozone tendency is the convergence of odd-oxygen mixing ratio products (the first right-

hand side term of Eq. 1). The total physics product (second term) includes the parameterized production and loss terms, and the analysis product (third term) is the corrected ozone tendency from data analysis. In MERRA2, the total ozone tendency from physics is decomposed into contributions from the chemistry, turbulence, and moist physics. Given a parameterized ozone



chemistry in MERRA2, the total ozone tendency from chemistry (CHM) is analyzed together with the correcting tendency term (i.e., CHM+ANA). The contributions of the turbulence and moist physics are negligible in the stratosphere (not shown) and therefore, are not considered in this analysis. We confirm that there are no major differences between the MERRA2 and AURA MLS ozone concentrations in the polar lower-to-mid stratosphere (see Appendix A1).

5    In addition, daily three-dimensional geopotential height, wind, and temperature fields from MERRA2 with the same period as the ozone were also employed in this study. We note that the nature of downward wave pulses and the associated wave geometry in MERRA2 were found to be in good agreement with the results from the European Centre for Medium-Range Weather Forecasts (ECMWF) Re-Analysis (Lubis et al., 2016a) and the NCEP/NCAR Reanalysis datasets (not shown) and are therefore robust among the various reanalysis products.

## 2.2    CESM1(WACCM) Simulation

The model simulation used in this study was performed with the NCAR's Community Earth System Model (CESM) version 1.0.2. CESM is a state-of-the-art coupled model system which includes an ocean, land, sea ice and atmosphere components (Hurrell et al., 2013). The atmospheric component of the CESM used in this study is the Whole Atmosphere Community Climate Model (WACCM), version 4 (Marsh et al., 2013), a fully interactive chemistry climate model consisting of a finite-

volume dynamical core with 66 standard vertical levels (up to 140 km or $\sim 5.1 \times 10^{-6}$ hPa) and a horizontal resolution of $1.9^{\circ}$ latitude $\times$ $2.5^{\circ}$ longitude. Interactive chemistry is calculated within the 3-D chemical transport Model of Ozone and Related Chemical Tracers, Version 3 (MOZART-3; Kinnison et al. 2007).

In this study one 100-yr simulation (1955-2054) is free run with fixed surface emissions of greenhouse gases (GHGs) and ozone depleting substance (ODSs) at 1960s levels, allowing us to study the ozone variability unmasked from any anthropogenic

influence. The simulation is run with interactive ocean and sea ice components. To represent a more realistic interaction between the tropics and extra-tropical dynamics, the QBO is nudged by relaxing tropical stratospheric zonal-mean winds towards observations following Matthes et al. (2010). The solar cycle is prescribed as spectrally resolved daily variations following Lean et al. (2005). Observed volcanic eruptions of the twentieth century are included.

### 2.3    Dynamical Diagnostics

The influence of eddies on ozone transport is quantified from the transform Eulerian mean (TEM) continuity equation for zonal mean tracer concentration (Andrews et al. 1987). For linear, steady, conservative waves, the complete Eulerian mean tracer transport equation can be written in the form:

$$\overline{\chi}_t = -\overline{v}^*\overline{\chi}_y - \overline{w}^*\overline{\chi}_z + e^{z/H}(\nabla \cdot \mathbf{M}) + \overline{S} \tag{2}$$

The Eq. 2 separates the local change in tracer concentration $\overline{\chi}_t$ as a result of transport processes that occur due to advection by

the residual circulation ($\overline{v}^*, \overline{w}^*$), the eddy effects ($\rho_0^{-1}\nabla \cdot \mathbf{M}$), and the chemical production minus loss rate ($\overline{S}$) (Andrews et al.,





1987). The eddy effects are defined as the divergence of the eddy transport vector ($\mathbf{M}$), with components defined as:

$$\mathbf{M} = \begin{pmatrix} \mathbf{M}^{(y)} \\ \\ \mathbf{M}^{(z)} \end{pmatrix} = \begin{pmatrix} -e^{-z/H}(\overline{v'\chi'} - \overline{v'\theta'}\,\overline{\chi}_z/\overline{\theta}_z) \\ \\ -e^{-z/H}(\overline{w'\chi'} + \overline{v'\theta'}\,\overline{\chi}_y/\overline{\theta}_z) \end{pmatrix} \tag{3}$$

The overbars indicate zonal means, primes are deviations from it and subscripts denote partial derivatives. The $\overline{v}^*$ and $\overline{w}^*$ in Eq. 2 denote the TEM residual meridional and vertical winds defined as $\overline{v}^* = \overline{v} - \rho_0^{-1}(\rho_0\overline{v'\theta'}/\overline{\theta}_z)_z$ and $\overline{w}^* = \overline{w} + (a\cos\phi)^{-1}$
$(\cos\phi\overline{v'\theta'}/\overline{\theta}_z)_\phi$, respectively. The potential temperature and scale height are represented by $\theta$ and $H$, respectively. The Coriolis parameter and Earth's radius are denoted by $f$ and $a$. In addition, the residual mean circulation calculated from the residual mass streamfunction ($\Psi_m$) and the wave geometry diagnostic are also used in this study (see Appendix B, Eq. B1-3).

### 2.3.1  Identification of DWC Event

We use a similar definition of DWC event as in Lubis et al. (2016a), which is based on daily total negative wave-1 meridional
heat flux ($\overline{v'T'}_{k=1}$) at 50 hPa averaged between 60° and 90°N, below the 5th percentile of the daily $\overline{v'T'}_{k=1}$ distribution. We focus our analysis on January, February and March (JFM), which is the period of maximum planetary wave coupling between the troposphere and stratosphere in the NH (Shaw et al. 2010; Lubis et al. 2016a). For time-lagged composites, the central date is the day on which $\overline{v'T'}_{k=1}$ has a minimum. Finally, the central dates of each event must be separated by at least 15 days. The time separation threshold is motivated by the time scale of planetary wave coupling between the stratosphere and troposphere
(Perlwitz and Harnik, 2003).

Applying our identification algorithm leads a total number of 19 potential DWC events in MERRA2 reanalysis (see Table 1) and 58 in CESM1(WACCM) (see Table S1 in supplemental material). The number of events found in MERRA2 reanalysis are similar as those reported by Dunn-Sigouin and Shaw (2015) using the ERA Interim reanalysis. We also note that the evolution of the downward and upward wave pulses in MERRA2 are in a good agreement with the results of Dunn-Sigouin
and Shaw (2015) (see Fig. S1). Moreover, it can be seen that potential DWC events are instantaneously linked to a transient reduction in ozone tendency in the Arctic stratosphere (see Table 1 for MERRA2 and Table S1 in supplemental material for CESM1(WACCM)). All the statistical significance for the composites is reported at the 95% level, based on a Monte Carlo approach following Lubis et al. (2016a). The composites are done both for the total fields and the anomalies, which are defined as deviation from the daily climatological seasonal cycle.

## 3  Direct Effects of the DWC on Polar Stratospheric Ozone

In the following sections we examine the effect of DWC events on polar stratospheric ozone in the observation and then in the model.



## 3.1 Observed Effects of DWC on Ozone

To examine the effects of DWC events on polar stratospheric ozone we first examine the connection between DWC, residual circulation, and Arctic temperatures during the composite life cycle in MERRA2 reanalysis. Figure 1 shows the evolution of the high-latitude residual circulation anomaly (a) and total potential temperature tendency (b) during the lifecycle of DWC as a function of pressure and time from -20 to +20 days. There is a positive (i.e., poleward) residual circulation anomaly and the corresponding positive potential temperature tendency in the stratosphere from days -15 to -5 (Figs. 1a,b). This positive potential temperature tendency is associated with air being advected downward over the pole, producing adiabatic warming. This behavior is consistent with anomalous EP flux convergence in the stratosphere induced by upward propagation of planetary-scale waves prior to DWC event (see Fig. S1a-b).

From days -3 to +3, the residual circulation anomaly and the potential temperature tendency subsequently change sign and reach their minimum values. The negative residual circulation anomaly suggests a deceleration of poleward transport of air mass, resulting in adiabatic cooling over this region. The negative residual circulation and temperature tendency are consistent with anomalous EP flux divergence in the stratosphere induced by transient downward wave propagation from the stratosphere to the troposphere during the maturation phase of DWC event (Fig. S1a-b).

Our results indicate that the life cycle of DWC involves a transient reversal of poleward to equatorward residual circulation anomaly, and subsequent changes in potential temperature tendency from positive to negative value. Therefore, it is possible that these effects could integrate to zero over the life cycles, meaning the impacts would be reversible. To check this, we calculated the time integration of both quantities averaged over the levels where their effects are significant. The results showed that both time-integrated residual circulation and potential temperature tendency over the composite of DWC life cycle is close to zero, indicating that the impacts is reversible (Figs. 1c-d). In addition, the results from the DWC lifecycle are compared with the results from the upward wave events. The upward wave event is defined in a similar manner as DWC event, but for the meridional heat flux values above the 95th percentile of the JFM distribution. It is shown that the time-integration of both quantities over the life cycle of upward wave events is positive, indicating a strengthening of residual circulation and net warming of the polar vortex (Figs. 1c-d and see also Fig. S2). The results indicate that the DWC event prevents a typical strengthening of residual circulation and the associated adiabatic warming induced by upward planetary waves.

Next, we analyze the implication of transient changes in residual circulation induced by DWC on polar stratospheric ozone in MERRA2. Figure 2 shows the corresponding transient evolution of the zonal-mean ozone tendencies averaged between 60 to 90°N. During downward wave events, the total ozone tendency transitions from a large positive value in the upper stratosphere to a large negative value in the lower stratosphere around day -4 (Fig. 2a). The evolution of the total ozone tendency is consistent with the evolution of high-latitude residual circulation anomalies (Fig. 1a). In particular, the positive (poleward) residual circulation anomaly in the stratosphere from days -15 to -5 leads to more ozone transport to the polar vortex, and the subsequent negative (equatorward) residual circulation anomalies between days -4 to +5 leads to less ozone transport to the polar vortex.



To investigate the source of the transient changes in polar stratospheric ozone during DWC events, we decompose the total ozone tendency (Fig. 2a) into contributions of dynamics and chemistry-plus-analysis terms. It is shown that the evolution of total ozone tendency in the mid-lower stratosphere is dominated by the dynamical term (Fig. 2b). The ozone tendency due to dynamics in the mid-lower stratosphere is mainly attributed to the ozone transport via vertical (advection) residual circulation

(Fig. 2e), while the tendency in the mid-upper stratosphere is mainly attributed to effects of eddy transport (Fig 2f). Therefore, the dominance of the dynamical term on the total ozone tendency in the mid-lower stratosphere during the composite lifecycle is consistent with the transient changes in residual mean transport (Fig. 1a). On the other hand, the contribution of the chemistry to the total ozone tendency (Fig. 2c) is evident in the upper stratosphere. While small significant regions of negative ozone tendency from chemistry are seen prior to the maturation phase of DWC (above 10 hPa days -10 to -5, Fig. 2c), the magnitudes

over the whole life cycle are relatively small compared to those caused by dynamics. These results suggest that transient changes in the polar mid-lower stratospheric ozone during the DWC life cycle are primarily due to changes in dynamical ozone transport.

The same conclusion can be drawn by assessing the instantaneous correlation between the two extreme stratospheric wave-1 heat flux events. Figure 3a shows a two dimensional histogram of total ozone tendency versus residual vertical wind anomaly

($w^*$) averaged over 60-90°N at 50 hPa in MERRA2. The black contour lines indicate the distribution of all daily JFM samples from 1980 - 2013 (90 days $yr^{-1}$ x 34 yr = 3060 days). The red and blue dots indicate the days with positive and negative extremes in total wave-1 heat flux, respectively (the top and bottom 5%). Negative extremes are associated with DWC events. The overall circular pattern of contours is evidence of a strong negative correlation between polar cap-averaged ozone tendency and $w^*$. This is consistent with a direct calculation of time series correlation, which is statistically significant (R=-0.81). In

addition, the days with positive and negative extremes (red and blue dots, respectively) are systematically skewed compared to the background distribution, suggesting that enhanced extreme negative heat flux (i.e., stronger DWC event) corresponds to a weaker residual circulation and a higher negative ozone tendency, and vice versa for positive extremes. In addition, Fig. 3b shows a similar diagnostic for the dynamical ozone tendency versus $w^*$. Again, the overall two dimensional distribution suggests a strong negative correlation between dynamical ozone tendency and the vertical component of the residual circulation,

with a temporal correlation coefficient of -0.82. This reflects the fact that the dynamical ozone tendency is strongly correlated with changes in residual circulation. We found no instantaneous relationship between chemical ozone tendency and $w^*$ (R=-0.17). Although the analyses in Fig. 3 are based on the 5% negative extreme events, the described relationships do not depend on the fraction of extreme events considered (not shown), and thus these results are representative of the general behavior. By using total column ozone tendency (TCO) instead of an ozone-mixing ratio at 50 hPa, the same conclusions are also obtained

and are even more robust (not shown).

The results showed that the impact of DWC events on ozone are transient and involve a positive to negative total ozone tendency evolution. Therefore, it is worth checking whether the effects on ozone could integrate to zero over their life cycles, indicating the impacts would be reversible. To examine this, we calculated the evolution of ozone tendency at 50 hPa averaged between 60 to 90°N and total column ozone (TCO) tendency for both DWC events (blue lines) and upward wave events (red

lines) (Fig. 4). The results showed that the time integration of the ozone and TCO tendencies over the life cycles of DWC is





close to zero, indicating a reversible impact on ozone (Fig. 4, horizontal blue lines). This result is consistent with the reversible impact of DWC on residual circulation (Fig. 1c). On the other hand, the time integration of the ozone and TCO tendencies during upward wave events is positive, indicating a net-increased ozone during the life cycle of upward wave event (Fig. 4, horizontal red lines and see also Fig. S3). This is again consistent with the irreversible poleward residual circulation during

upward planetary wave events (Fig. 1c). In summary, these results suggest that the direct effect of full wave reflection life cycle is to prevent the typical increase of ozone due to upward planetary wave events.

## 3.2 Simulations of the Effects of DWC on Ozone

Determining the connection between DWC, stratospheric residual circulation, and polar temperature is one of the keys to improving our understanding of the link between stratospheric dynamics and ozone variability, both in the real atmosphere and

10 in the stratosphere-resolving chemistry-climate models. In this section, we attempt to test if the linkages between DWC and transient changes in polar stratospheric ozone can be reproduced in a current chemistry-climate model, CESM1(WACCM).

Figure 5 shows the evolution of residual circulation anomaly and potential temperature tendency during DWC event in CESM1(WACCM). Consistent with MERRA2, the model simulation showed that the life cycle of DWC events involve a transient reversal of residual circulation anomalies and potential temperature tendencies. In particular, there is a positive residual

circulation anomaly (Fig. 5a), and the corresponding positive potential temperature tendency in the Arctic stratosphere (Fig. 5d) from days -20 to -6. This is consistent with an anomalous EP flux convergence in the stratosphere induced by upward propagating waves prior to the maturation phase of DWC event (not shown). From days -3 to +3, both residual circulation anomaly and temperature tendency subsequently change sign and reach its minimum value. The equatorward anomaly of the residual circulation and the associated adiabatic cooling in the stratosphere is again consistent with wave-1 EP flux divergence

in the stratosphere due to transient downward wave propagation from the stratosphere to the troposphere (not shown).

Finally, the time integration of the residual circulation anomaly and polar temperature tendency during the composite life-cycle was analyzed in Figs. 5c-d. It is shown that the impacts on both residual circulation and potential temperature tendency over the life cycle of DWC events are reversible. This result supports the observational-based analysis with MERRA2, that the DWC event acts to dampen the typical strengthening of residual circulation and the associated adiabatic warming induced by

25 upward planetary waves.

Next we examine the effect on ozone tendencies, as was done in Fig. 2. Figure 6 shows the time evolution of total ozone tendencies during the composite life cycle. Consistent with MERRA2, the negative ozone tendency in the model occurs during the time of strongest DWC events (days -3 to +3), between 100 and 5 hPa (Fig. 6a). This negative ozone tendency is preceded by a positive ozone tendency (days -20 to +5). The transition of positive to negative ozone tendency in the lower-to-mid

stratosphere is consistent with poleward to equatorward residual circulation anomalies (Fig. 5a). Interestingly, there is a strong positive ozone tendency in the upper stratosphere (above 5 hPa) during the time of maximum DWC events, which is not captured by MERRA2. This discrepancy will be discussed in the following paragraph.

By decomposing the total ozone tendency into dynamical and chemical terms, it is shown that transient changes in ozone dynamics dominate the total ozone tendency during the composite life cycle (Fig. 6b-c). In the mid-lower stratosphere, the





ozone tendency due to dynamics is mainly attributed to the vertical advection process (Fig. 6d), while in the upper stratosphere the eddy transport effect becomes dominant (Fig. 6f). In addition, the strong positive total ozone tendency in the upper stratosphere during the time of maximum DWC events is attributed to the dynamical ozone tendency due to eddy transport effect and vertical advection through the residual circulation (Fig. 6f). The magnitude of these two quantities in the upper stratosphere

is relatively higher in the model compared to MERRA2, and therefore, leads to the differences in ozone tendency in the upper stratosphere (Fig. 2d,f vs. Fig. 6d,f). On the other hand, the ozone tendency due to chemistry in the upper stratosphere is somewhat weaker and in the opposite sign of the ozone tendency due to dynamics (Fig. 6c). The overall results support our observational-based analysis that the transient changes in polar stratospheric ozone during a DWC event are mainly attributed to changes in the dynamical ozone transport.

As in MERRA2, this general relationship between DWC and polar stratospheric ozone in the model simulation can be also assessed through the instantaneous correlation between the two extreme stratospheric wave-1 heat flux events. Figure 7 shows a similar diagnostic as in Fig. 3, comparing the two-dimensional distribution of the polar cap-averaged ozone tendencies and residual circulation during extreme wave-1 heat flux events. Consistent with MERRA2, the days with positive and negative extremes (red and blue dots, respectively) are systematically skewed compared to the background distribution, so that stronger

DWC events correspond to stronger negative ozone tendency and weaker residual circulation (Figs. 7a). A similar instantaneous correlation is found between dynamical ozone tendency and w*, but not for the ozone tendency due to chemistry. This is again consistent with the observational-based analysis, that the polar mid-lower stratospheric ozone during the DWC life cycle is instantaneously linked to changes in dynamical ozone transport.

   As a last step, the direct cumulative effects of DWC events on ozone in the model were examined in the same way as in Fig.

4. The time evolution of total column ozone (TCO) tendency and ozone tendency at 50 hPa averaged between 60 to 90°N for the composite of DWC event (blue lines) and upward wave event (red lines) is shown in Fig. 8. Confirming the results from MERRA2, the time integration of significant TCO and ozone tendencies is zero, indicating a reversible impact on the ozone (Fig. 8, blue lines). On the other hand, the time integration of the significant TCO and ozone tendencies during upward wave event are positive, indicating a net increase in ozone (Fig. 8, red lines). This is consistent with our previous findings suggesting

that the direct effect of the full wave reflection life cycle is to prevent the typical increase of ozone due to upward planetary wave events.

## 4 Seasonal Impact of DWC on Ozone

The former analysis shows that an individual DWC event has a statistically significant impact on the polar stratospheric ozone. While the impact of an individual event occurs on a short time scale, several events in an individual JFM season may produce

an impact on a longer time scale. In this section, we briefly examine the cumulative impacts of DWC on ozone during seasons that are dominated by DWC events.





## 4.1 Reflective versus Absorptive Winters

In order to analyze the seasonal impact[1] of DWC on polar stratospheric ozones, we classify winters as reflective and absorptive based on basic state of the stratosphere during winter. The classifications are based on the vertical wave numbers ($m$) (averaged between 1-5 hPa and 65-75ºN) and zonal mean wind ($U$) at 30 hPa (averaged between 60-85ºN) in JFM, similar to Perlwitz
and Harnik (2003).

Reflective winters (with dominant DWC events) are defined when $m < 0$ and $U(30) > 2.5\sigma$, while absorptive winters (with dominant upward wave events) are defined when $m > 0$ and $U(30) < 2.5\sigma$. We exclude the years with (without) SSW events from the potential reflective (absorptive) winters. The time series of selected reflective and absorptive winters are shown in Fig. 9. Using this definition, we found the most reflective (absorptive) winters, 8 (11) in MERRA and 30 (29) in CESM1(WACCM)
(see Table 2). We also note that the composites of total vertical components of wave-1 EP flux during reflective (absorptive) winters are negative (positive) in the lower-to-mid stratosphere, which is consistent with downward (upward) wave propagation (not shown).

Figure 10 shows composites of the zonal-mean wind, wave-1 geopotential height, and temperature difference in JFM for the composite of reflective and absorptive winters in MERRA2. During reflective winters, the maximum zonal-mean zonal
wind resides in the mid-stratosphere, and consequently the region of vertical reflecting surface extends down to 3 hPa (Fig. 10a). This vortex configuration is favorable for DWC events, which is indicated by an eastward phase tilt with heights of the wave-1 structure from the mid-troposphere to the mid-stratosphere (Fig. 10b). In addition, the Arctic mid-lower stratosphere is significantly colder; the polar cap temperature at 50 hPa is 7 K lower than the climatological mean (Fig. 10c). This cooling is consistent with a strong and stable polar vortex, which is associated with less wave absorption in the stratosphere due to
enhanced DWC events.

In contrast, during absorptive winters, the vertical reflecting surface shifts upward into the upper stratosphere and the meridional waveguide becomes wider in the mid-stratosphere (Fig. 10d). This configuration is favorable for upward wave events, indicated by the westward phase tilt with height of the wave-1 structure (Fig. 10e). The Arctic mid-lower stratosphere is significantly warmer, by approximately 6 K higher than the climatological mean at 50 hPa (Fig. 10f). The warmer and more disturbed
polar vortex during absorptive winters is consistent with enhanced upward propagating waves from the troposphere into the stratosphere, resulting in stronger wave-mean flow interaction.

To check the robustness of the results, we repeated the analysis with a 100-yr CESM1(WACCM) simulation (Fig. 11). In general, the structures of the vortex, wave geometry, vertical wave-1 pattern, and Arctic temperature anomalies are in agreement with the MERRA2 (Fig 11). The reflective winters are characterized by a stronger vortex, a pronounced vertical reflecting
surface, an eastward phase tilt of wave-1 pattern in the lower-to-mid stratosphere (i.e., dominated by DWC events), and a significantly colder stratosphere (by approximately 7 K at 50 hPa). On the other hand, the absorptive winters are characterized

---

[1]Previous studies attempted to use cumulative wave-1 heat flux values in JFM (e.g., Shaw and Perlwitz 2013) to analyze the seasonal impact of DWC. However, we find this method comes with caveats. For example, the lower cumulative wave-1 heat flux values in JFM can be attributed to increased wave reflection events after SSW events (Kodera et al., 2008). Therefore, 4 out of 8 most active DWC seasons defined by Shaw and Perlwitz (2013) (see their Fig. 10) are associated with weaker vortex condition (i.e., absorptive state).



by a weaker vortex, an absence of vertical reflecting surface, a westward phase tilt of wave-1 pattern (dominated by upward propagating waves), and a significantly warmer stratosphere (by approximately 5 K at 50 hPa).

## 4.2 Seasonal Impact on Ozone in Winter and Spring

In order to estimate the seasonal impacts of DWC events on ozone, we analyzed ozone differences between reflective and absorptive seasons for mid winter (Jan-Feb) and late winter/early spring (Mar-Apr) in the MERRA2 (Fig. 12) and in the model (Fig. 13). Here, we call the reflective seasons (i.e., seasons dominated by DWC events) as REF, and the absorptive seasons(i.e., seasons dominated by upward wave events) as ABS.

It is shown that the seasonal effects of DWC leads to a reduction of ozone concentration in the stratosphere during mid winter and early spring (Figs. 12a,d and Figs. 13a,d). These results are in agreement with our analysis over the life cycle of DWC showing that: (1) a reduction of poleward residual circulation induced by DWC event leads to less ozone transport to the pole during mid winter (the direct effect of DWC) and (2) the effect of DWC on polar temperature leads to the cold polar vortex in mid winter, and thus, resulting in more ozone loss during early spring (the indirect effect of DWC). To distinguish between the two effects quantitatively, we examine the contribution of dynamics and chemistry terms to the mean differences between the two types of winters. It can be seen that a reduction of ozone concentration during mid winter in REF compared to ABS is maintained by negative ozone tendency due to dynamics in both MERRA2 (Fig. 12b) and model simulation (Fig. 13b). This is consistent with the direct impact of DWC on ozone in mid winter, which results in weaker poleward transport of ozone to the pole. Comparing the ozone tendency due to dynamics in REF to the climatological mean values (see green contour lines in Fig. 12b and Fig. 13b), we found that the accumulative impact of DWC in mid winter reduces the seasonally averaged dynamical ozone transport to the polar region (not shown).

On the other hand, during early spring, the lower ozone concentration in REF compared to ABS in the upper stratosphere (between 5 and 1 hPa) and in the lower-to-mid stratosphere (between 100 hPa and 10 hPa) (Fig. 12d and Fig. 13d) is maintained by negative ozone tendency due to chemistry terms (Fig. 12f and Fig. 13f). This is consistent with the indirect impact of DWC on ozone that leaves the polar vortex cold in mid winter (Fig. 1d, Fig. 5d, Fig. 11c, and Fig.12c), and thus resulting in more ozone destruction in spring due to more accumulation of ODS on polar stratospheric clouds. This is also consistent with a strong polar vortex associated with increased DWC events (Figs. 10a and 11a). Comparing the ozone tendency due to chemistry in REF to the climatological mean values (see green contour lines in Fig. 12f and Fig. 13f), we found that the indirect accumulative impact of DWC in spring results in increase of seasonally averaged chemical ozone loss (not shown). The aforementioned indirect effect of DWC should only affect ozone concentration in spring when the polar stratosphere becomes sunlit. Moreover, it is also shown that the dynamical ozone tendency during this season is stronger in REF compared to ABS (Fig. 12e and Fig. 13e). This behavior is most likely associated with the sharpening of meridional and vertical gradients of ozone after the spring time ozone loss occurs.





## 5 Conclusions and Discussion

The goal of this study was to investigate the impact of DWC on polar stratospheric ozone in order to fully understand the mechanisms controlling the variability of Arctic stratospheric ozone. The key results of this study are as follows:

1. The life cycle of DWC events involves a transient reversal of poleward to equatorward residual circulation anomalies, and subsequent changes in potential temperature tendency. The impact on residual circulation and potential temperature tendency are reversible over the lifecycle, indicating that DWC prevents a typical strengthening of residual circulation and the associated adiabatic warming induced by upward planetary wave events.

2. Transient changes in polar stratospheric ozone during the life cycle of DWC events are primarily attributed to changes in the dynamical ozone transport.

3. The direct effect of DWC events on ozone is reversible, indicating that DWC prevents a typical increase of ozone due to upward planetary wave events. This is consistent with a reversible impact of DWC on the residual circulation in winter.

4. The indirect effect of DWC events leads to increased springtime ozone loss. This is consistent with the reversible impact of DWC on polar temperature that leads to a cold polar vortex in winter (i.e., DWC prevents the typical warming induced by upward planetary wave events) and thus, resulting in larger springtime chemical ozone loss.

5. Winter seasons dominated by DWC events (i.e., reflective winters) are characterized with a lower stratospheric ozone concentration in winter and spring. This behavior is consistent with the cumulative effects of downward planetary wave events on polar stratospheric ozone via the residual circulation and the polar temperature in winter.

Our results establish a new perspective on dynamical processes controlling Arctic ozone variability. Previous studies have shown that variations in upward planetary waves entering the lower stratosphere in midwinter determine the magnitude of ozone loss in the Arctic polar vortex (e.g., Randel et al. 2002; Newman et al. 2001; Weber et al. 2003; Tegtmeier et al. 2008). In particular, these studies showed that weaker midwinter planetary wave forcing in the stratosphere due to weaker upward wave propagation leads to lower spring Arctic temperatures, and thus to more ozone destruction in spring. Our results suggest that weaker midwinter planetary wave forcing in the stratosphere can also be attributed to enhanced DWC events (instead of less tropospheric wave source) in the presence of the vertical reflecting surface. Our proposed mechanism is schematically shown in Fig. 14a. Increased DWC events in the presence of the vertical reflecting surface lead to a positive divergence of the EP flux in the stratosphere, resulting in less ozone transport to the pole due to a weaker residual circulation in winter. Furthermore, the cold polar vortex induced by increased DWC events, leads to more ozone destruction in spring via heterogeneous processes. In contrast, if the winters are dominated by upward wave events as shown in Fig. 14b, the transport of ozone to the pole will increase due to large EP flux convergence (red shading) induced by upward wave propagations. This condition can lead to less springtime ozone loss due to a warm polar vortex. Our results suggest that a larger springtime ozone loss can occur if the DWC events dominate the upward wave events in the Arctic winter stratosphere.



Recent studies have shown that large chemical ozone loss in the spring of 2011 is one of the major reasons for the unprecedented low Arctic column ozone (e.g., Manney et al. 2011; Isaksen et al. 2012; Hommel et al. 2014). This is attributed to extremely low mid-winter temperatures in the lower stratosphere resulting from weaker midwinter planetary wave forcing (Hurwitz et al., 2011). Shaw et al. (2014) hypothesized that the extreme negative eddy heat flux events observed during March 2011 could contribute to ozone loss via a weakening of the residual circulation, which leads to weakened transport, a lowering of Arctic temperatures and thus large springtime ozone loss. Our results confirmed this by showing that a lower ozone concentration in mid winter and early spring during reflective years are associated with both a weakening of ozone transport and an increase of springtime ozone loss induced by DWC events. Since winter of 2011 is classified as a reflective winter – characterized by a vertical reflecting surface and strong downward wave reflection–, the lower Arctic column ozone in 2011 can thus be attributed also to enhanced DWC events.

Our results also reveal that the amount of wave absorption directly influences polar Arctic temperatures, and therefore the amount of ozone destruction in spring. Since wave absorption is minimal during reflective winters, the winters tend to be cold with more ozone destruction. The process of wave reflection and absorption is highly variable, and the amount and location depends on the tropospheric source of the waves, on the structure of the vortex, on which the waves propagate, and on non-conservative effects (McIntyre and Palmer, 1983; Harnik and Lindzen, 2001; Harnik and Heifetz, 2007). Therefore, a better understanding of the conditions that lead to such events is needed to improve our understanding of the link between stratospheric dynamics and ozone variability.

A recent multi-model inter-comparison of chemistry climate models (CCMs) concludes that the models do not produce a consistent prediction of the evolution of Arctic temperatures and ozone loss in the twenty-first century, mainly because of discrepancies in the model's dynamics (SPARC CCMVal 2010, Chapter 4). Understanding the impact of DWC events on polar stratospheric ozone and temperatures could provide a useful diagnostic to validate the influence of stratospheric dynamics on springtime column ozone in coupled CCMs.

## Appendix A: Ozone Profiles in MERRA2

The Aura Microwave Limb Sounder (MLS) ozone product (Waters et al., 2006) from 2005 to 2013 was used to validate the quality of ozone data from MERRA2. Figures 1a and 1b show the vertical profiles of zonal mean ozone mixing ratios in MERRA2 and MLS averaged between 60-90ºN for (left) annual and (right) winter JFM means, from 2005 and 2013. It can be seen that the vertical profile of ozone mixing ratios, both mean (contour) and ranges (shading) in MERRA2, shows a reasonable agreement with MLS data from the lower to midstratosphere. Above 3 hPa – where photochemical processes become more dominant –, both the mean and the spread in MERRA2 slightly deviate from the MLS, which is likely associated with model ozone biases in MERRA2 due to a simplified (parameterized) ozone chemistry used in GEOS-5 model (Bosilovich et al., 2015). Furthermore, we also confirm that there are no major differences between the MERRA2 and MLS total column ozone climatologies (Fig. A1c). As shown by the solid red line and spreads in Fig. 1c, the MERRA2 total column ozone differs very little from MLS.





Figure A2 shows the differences between the daily total ozone and the ozone anomaly time series in MERRA1 and MERRA2 relative to MLS from 2005 to 2013. It is shown that a positive bias of total ozone mixing ratio in MERRA1 in the lower-to-mid stratosphere (between 100 and 5 hPa) is no longer exists in MERRA2 (Figs. A2d,e). There is also good improvement in total ozone mixing ratio in the upper stratosphere, as indicated by a significant reduction of negative ozone bias in MERRA2

in comparison to MERRA1. Likewise, ozone anomaly in MERRA2 is in a good agreement with MLS dataset compared to MERRA1 (Figs. A2i,j). In summary, we find that the ozone dataset from MERRA2 is significantly improved compared to MERRA1 (Rienecker et al., 2011) (see Figs. A2) and thus, the analysis based on this dataset is meaningful and more reliable.

## Appendix B:  Residual-Mean Circulation and Wave Geometry Diagnostics

Understanding how circulation is controlled by planetary waves is the key to connecting the wave driving to polar temperatures

and transport of trace gases such as ozone. The transformed Eulerian mean formulation (Andrews et al., 1987) can be used to directly examine wave effects on the circulation and ozone transport to the polar vortex (Fusco and Salby 1999; Plumb 2002; Lubis et al. 2016b). The residual mean circulation is calculated from the streamfunction ($\Psi$) of the residual TEM meridional $\overline{v}^*$ and vertical $\overline{w}^*$ winds, as follow:

$$(\overline{v}^*, \overline{w}^*) = \frac{1}{\rho_o \cos\phi} \left( -\Psi_z, \frac{1}{a}\Psi_\phi \right), \tag{B1}$$

and hence,

$$\Psi^* = \rho_o \cos\phi \int\limits_z^\infty \overline{v}^* e^{-z/H} \partial z, \tag{B2}$$

The residual mass streamfunction $\Psi_m^\dagger$ ( units in kg/s) we obtain by:

$$\Psi_m^\dagger = (2\pi a)\, \rho_o \cos\phi \int\limits_z^\infty \overline{v}^* e^{-z/H} \partial z. \tag{B3}$$

where $\Psi_m^\dagger \to 0$ as $z \to \infty$.

Furthermore, in order to diagnose vertical reflecting surfaces and meridional wave guides for stratospheric planetary wave propagation, the index of refraction ($n^2$) is separated into contribution from meridional and vertical wavenumbers using a quasi-geostrophic (QG) potential vorticity (PV) equation on a $\beta$ plane, with a specified zonal wavenumber ($k$) and phase speed ($c$), as follow (see Harnik and Lindzen 2001; Lubis et al. 2016a, for details):

$$\begin{aligned} n^2 &= \frac{N^2}{f^2}\left\{ \frac{\overline{q}_y}{\overline{u}-c} - k^2 + f^2\frac{e^{z/2H}}{N}\left( \frac{e^{-z/H}}{N^2}\left( e^{z/2H}N \right)_z \right)_z \right\} \\ &\equiv m^2 + \frac{N^2}{f^2}l^2. \end{aligned} \tag{B4}$$

where $\overline{q}_y$ is the meridional gradient of zonal mean potential vorticity, $N^2$ is the buoyancy frequency, and $\beta$ is the variation of the Coriolis parameter with latitude. Since the basic state is non-separable in latitude and height, this equation is solved





by using a QG model (Harnik and Lindzen, 2001), in which the wave numbers are diagnosed from the wave solution $\psi$, as $m^2 = -Re(\psi_{zz}/\psi)$ and $l^2 = -Re(\psi_{yy}/\psi)$. A reflecting surface for vertical propagation is the $m^2 = 0$ surface.

*Acknowledgements.* This work is supported by the German-Israeli Foundation for Scientific Research and Development under grant GIF1151-83.8/2011. This work has also been partially performed within the Helmholtz-University Young Investigators Group NATHAN funded by

5  the Helmholtz-Association through the Presidents Initiative and Networking Fund and the GEOMAR Helmholtz Centre for Ocean Research Kiel, Germany. NH also acknowledges the support of a Rossby Visiting Fellowship from the International Meteorological Institute (IMI) of Stockholm University, Sweden. We also thank Mijeong Park for providing daily AURA-MLS ozone dataset, and NASA's Global Modeling and Assimilation Office for providing the MERRA-1 and MERRA2 dataset. The model simulations were performed at the German Climate Computing Centre (Deutsches Klimarechenzentrum, DKRZ) Hamburg, Germany.



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





**Table 1.** Central dates of potential DWC events at 50 hPa and the minimum total 5-day smoothed wave-1 heat flux (K. m/s) and ozone tendency (x$10^{-2}$ ppmv/day) from 60 to 90$^o$N during the event in MERRA2.

| Dates | $\min_{60-90^o N} \overline{v'T'}_{k=1}$ | $\min_{60-90^o N} dO_3/dt$ (MERRA2) | $\min_{60-90^o N} dO_3/dt$ (MLS) |
|---|---|---|---|
| 17 Feb 1981 | -37.84 | -20.11 | - |
| 14 Mar 1982 | -68.71 | -12.62 | - |
| 25 Jan 1986 | -44.07 | -14.86 | - |
| 26 Feb 1989 | -120.38 | -29.83 | - |
| 14 Feb 1990 | -54.73 | -14.65 | - |
| 23 Mar 1990 | -53.08 | -18.74 | - |
| 1 Feb 1991 | -27.35 | -13.61 | - |
| 23 Jan 1992 | -50.77 | -10.15 | - |
| 22 Mar 1993 | -52.17 | -13.63 | - |
| 11 Mar 1994 | -40.07 | -8.21 | - |
| 28 Mar 1995 | -72.76 | -22.31 | - |
| 13 Jan 1996 | -51.95 | -8.62 | - |
| 14 Mar 1996 | -151.45 | -47.11 | - |
| 23 Mar 2000 | -31.86 | -18.49 | - |
| 08 Mar 2002 | -50.09 | -12.24 | - |
| 10 Jan 2007 | -39.52 | -7.51 | -6.90 |
| 21 Mar 2007 | -73.02 | -21.5 | -27.41 |
| 28 Jan 2008 | -65.16 | -4.33 | -3.58 |
| 28 Feb 2008 | -79.17 | -15.16 | -13.24 |





**Table 2.** The reflective and absorptive years defined based on the vertical reflecting surface ($m < 0$) and $U_{30hPa}$ in JFM.

| Data | Reflective | Absorptive |
|---|---|---|
| MERRA2 | 1982, 1986, 1990, 1993, 1994, 1995, 1997, 2011 | 1979, 1981, 1984, 1985, 1999, 2001, 2004, 2008, 2009, 2010, 2013 |
| CESM1(WACCM) | 1958, 1964, 1966, 1967, 1968, 1969, 1971, 1972, 1974, 1982, 1984, 1985, 1986, 1990, 1991, 1994, 1998, 2004, 2005, 2011, 2019, 2024, 2028, 2030, 2033, 2035, 2042, 2045, 2047, 2049 | 1956, 1957, 1960, 1965, 1970, 1975, 1978, 1979, 1980, 1981, 1988 , 1989, 1992, 1995, 2002, 2014, 2017, 2018, 2020, 2023, 2026, 2027, 2032, 2036, 2038, 2043, 2044, 2046, 2048, 2052, 2054 |



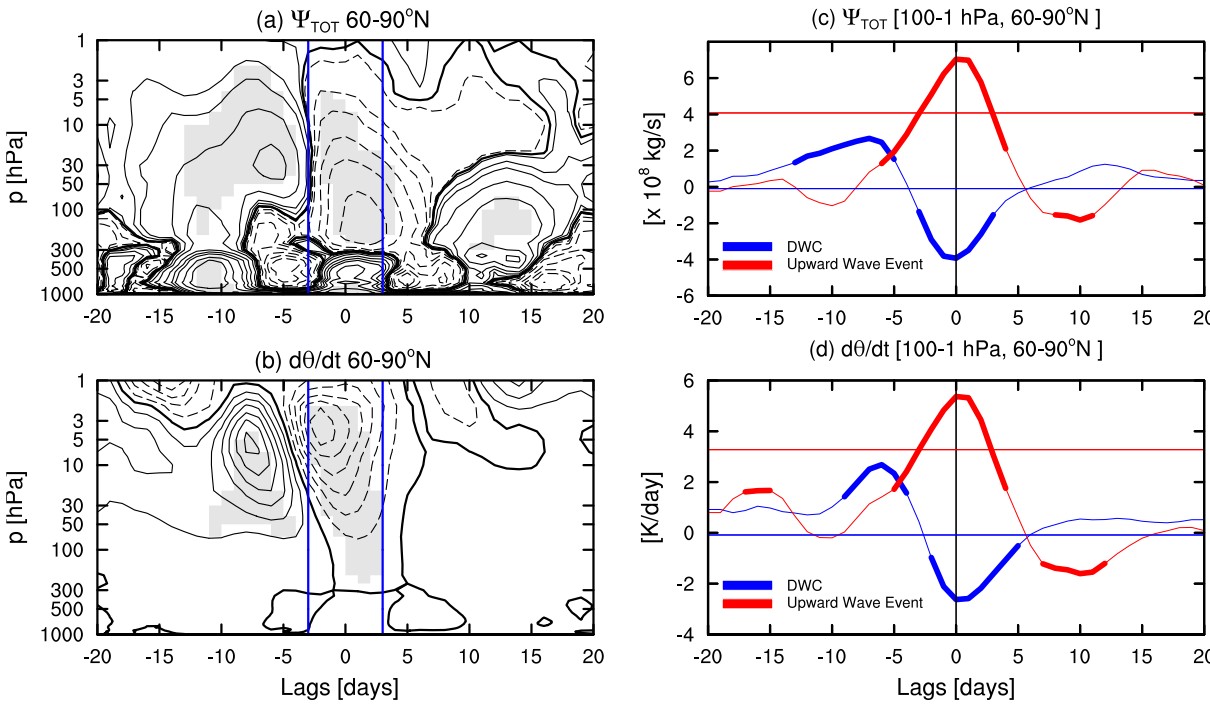

**Figure 1.** (a-d) Evolution of the downward planetary wave events as a function of time from days -20 to +20 and pressure: (a) residual mass-streamfunction anomaly, and (b) potential temperature tendency averaged from 60 to 90°N. The contour intervals are $\pm\ 1 \times 10^9$ [0.5, 1, 2, 4, 8, 16, 32, 64,..] kg s$^{-1}$ for Fig. 1a and $\pm\ 0.5$ K day$^{-1}$ for Fig. 1b. Shading indicates statistical significance at the 95% level. The periods of the maximum DWC event (days -3 to +3) are bounded by two vertical blue lines. (c-d) As in Figs. 1a-d, but both quantities are averaged in height from 100 to 1 hPa. The downward wave event is denoted by the blue line, while the upward wave event is denoted by the red line (further discussed in the text). The horizontal lines (blue and red) indicate the time-integrated significant values of each quantity over the life cycle. The time-integrated values for the upward wave event are divided by 10 for display purposes. Statistical significance at the 95% level is denoted with thick lines.





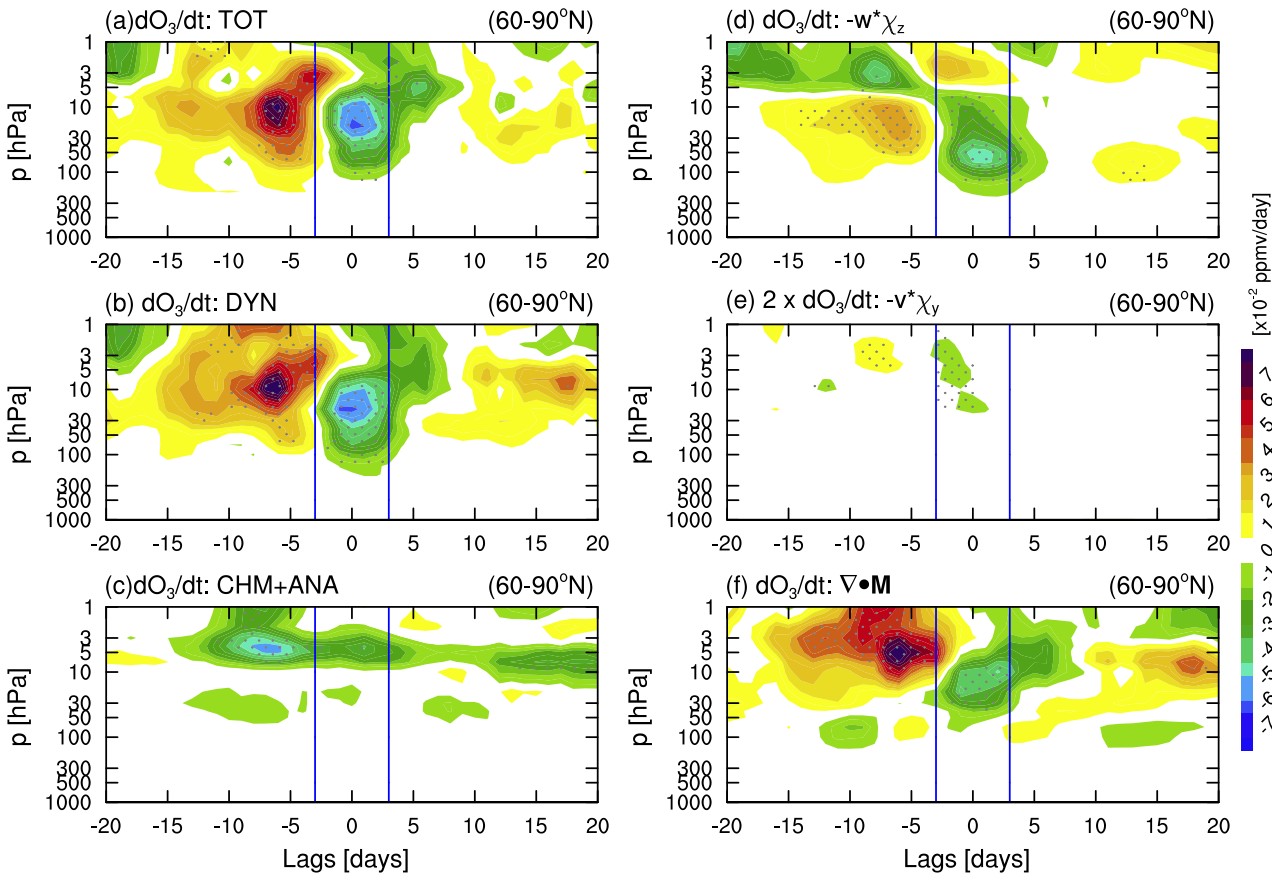

**Figure 2.** Evolution of the ozone tendencies for the composite DWC event as a function of time and pressure, averaged from 60 to 90°N: (a) total ozone tendency, (b) ozone tendency anomaly due to dynamics and (c) due to parameterized chemistry. Tendency from the dynamics is decomposed into (d) vertical advection, (e) meridional advection, and (f) eddy transport effects based on Eq. 1. The periods of the maximum DWC event (days -3 to +3) are bounded by two vertical blue lines. Stippling indicates statistical significance at the 95% level using a Monte Carlo approach.



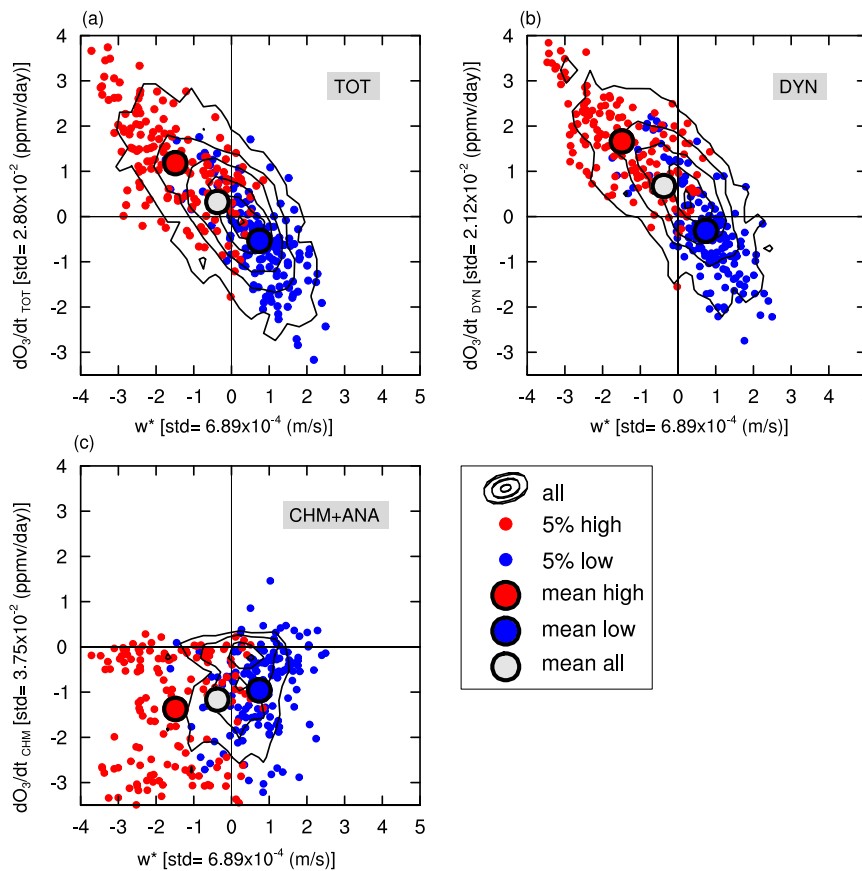

**Figure 3.** Two-dimensional joint probability density distribution of (a) total ozone tendency vs residual vertical wind velocity (averaged over 60 to 90°N at 50 hPa) in JFM, (b-c) as in Figure 3d, but for total ozone tendency due to dynamics and to parameterized chemistry, respectively. The axes are normalized to standard deviations in each quantity. Black contours show the probability distribution of all JFM days during 1979-2013 (contour shown are 1, 2, 3, 4, 5,.. % of total). Red and blue dots show days with 5% extreme maximum and minimum total wave-1 meridional heat flux. Larger circles indicate the mean of the distribution of all points (grey), high extremes (red), and low extremes (blue).





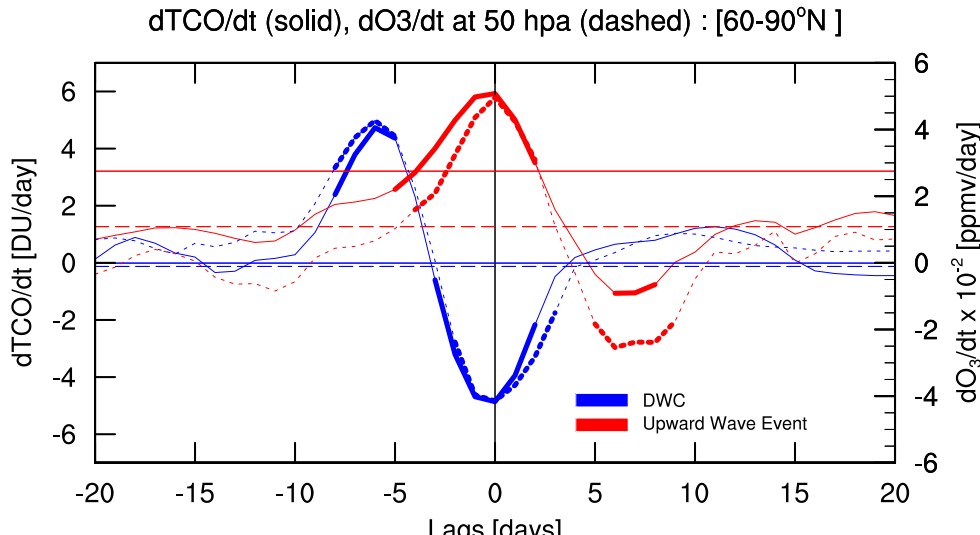

**Figure 4.** Evolution of the total column ozone tendency (solid lines) and mixing ratio ozone tendency (dashed lines) at 50 hPa, averaged between 60-90°N for DWC events (blue) and upward wave events (red). Statistical significance at the 95% level is denoted with thick lines based on a Monte Carlo approach.




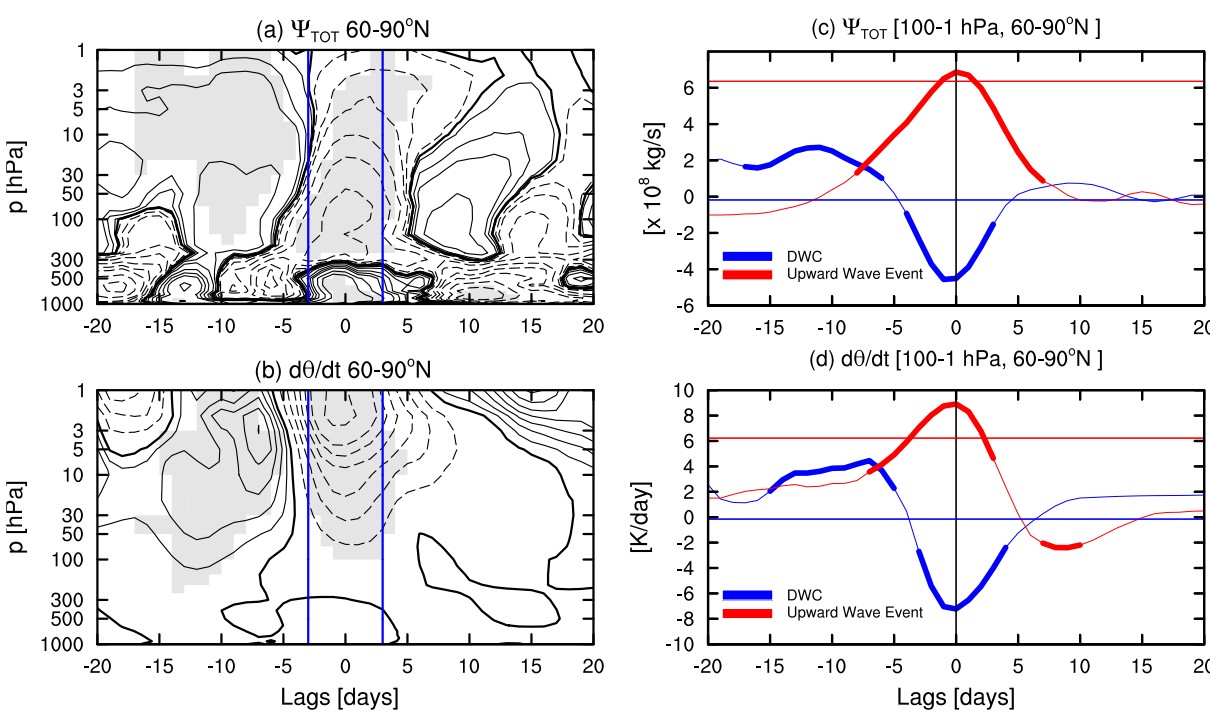

**Figure 5.** Same as Fig. 1, but from a 100-yr CESM1(WACCM) simulation.





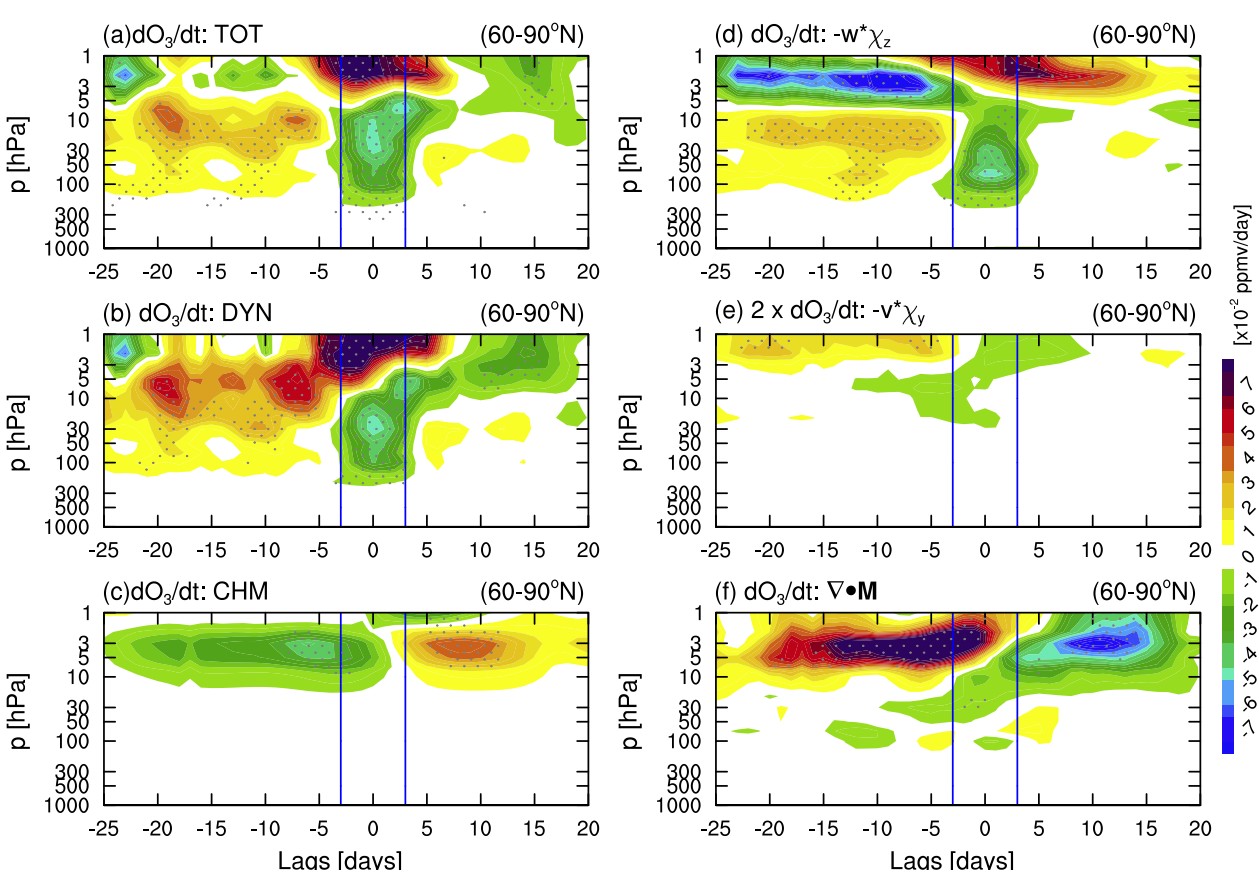

**Figure 6.** Same as Fig. 2, but from a 100-yr CESM1(WACCM) simulation.



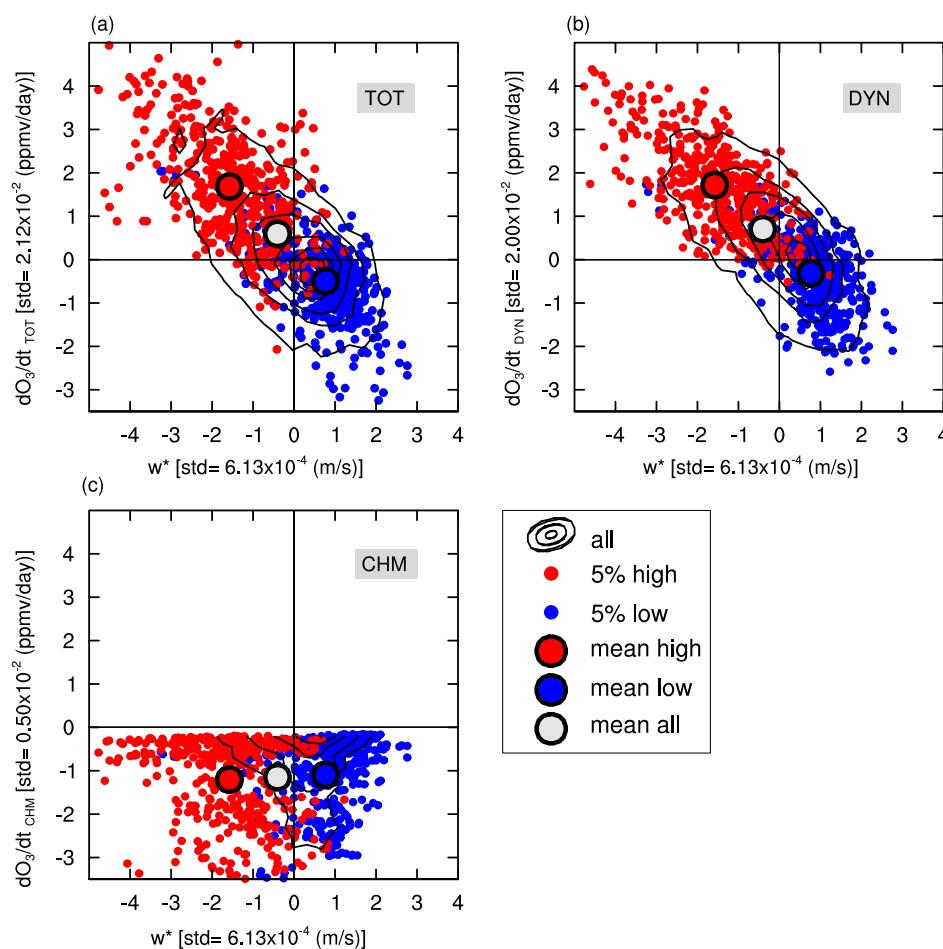

**Figure 7.** Same as Fig. 3, but from a 100-yr CESM1(WACCM) simulation.





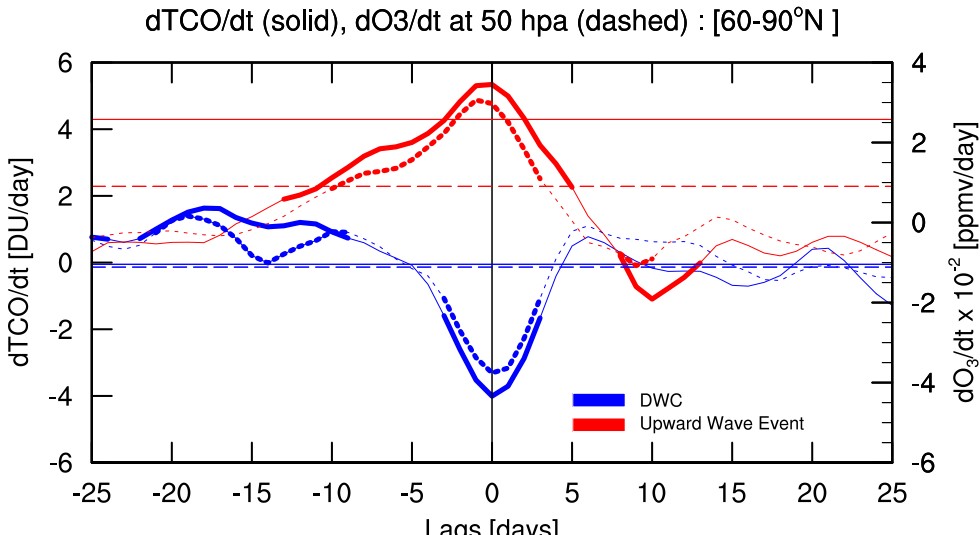

**Figure 8.** Same as Fig. 4, but from a 100-yr CESM1(WACCM) simulation.





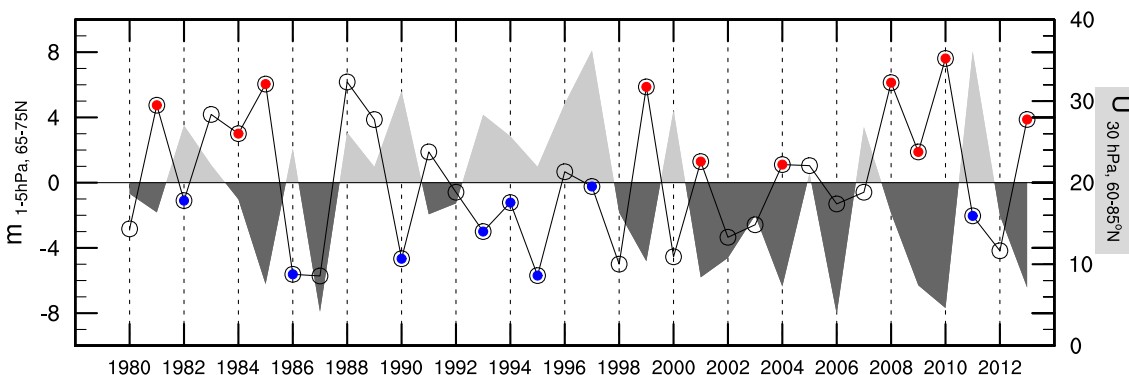

**Figure 9.** Time series of seasonal (JFM) mean of $m$ and $U(30)$. The blue (red) asterisk indicates the reflective (absorptive) winters defined based on vertical wavenumbers and mid-stratosphere zonal mean wind (further discussed in the text).



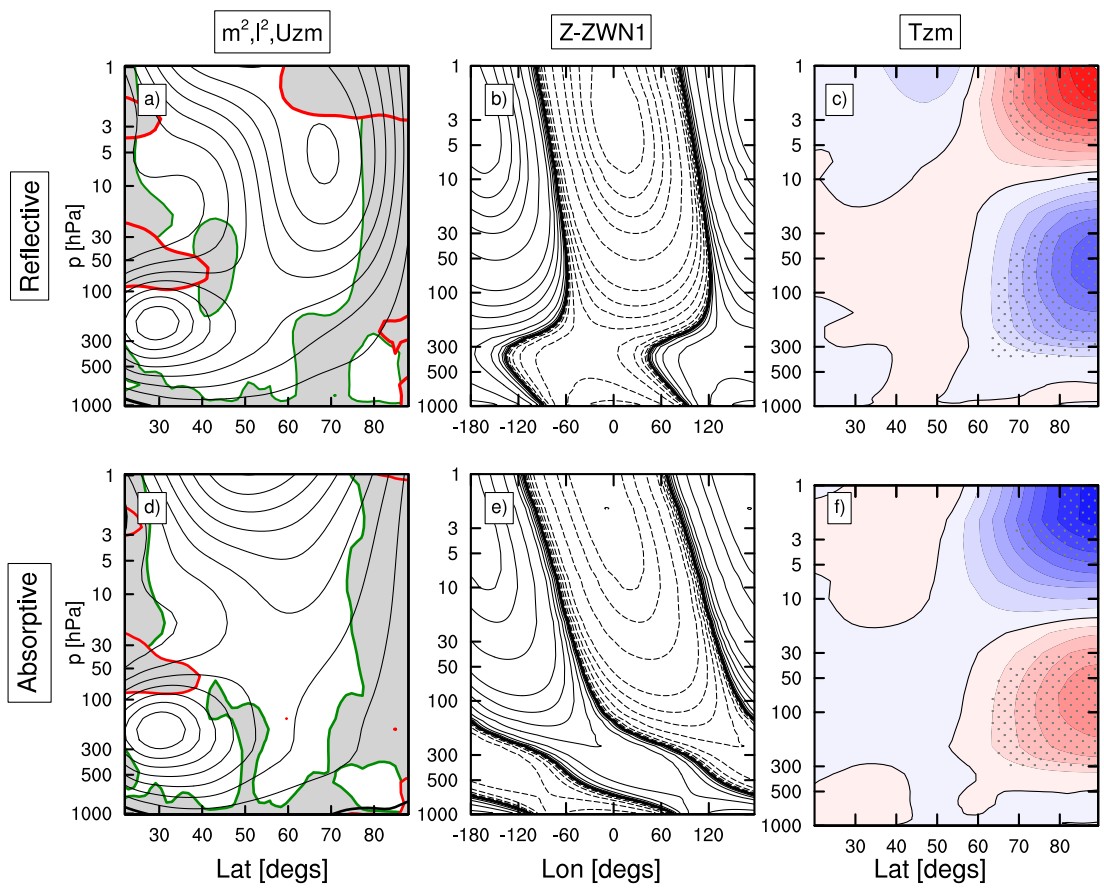

**Figure 10.** (a, d) Composites of the zonal wind and wave geometry for (top) reflective and (bottom) absorptive winters in JFM from MERRA. The grey shading indicates regions of wave evanescence and the red (green) contour lines indicates the vertical reflecting surface $m^2 = 0$ (meridional wave guide $l^2 = 0$); contour interval for the zonal wind is 5 m/s. (b-e) As in (a, d), but for composites of the wave-1 geopotential heights averaged between 60 to 70°N; contour interval is logarithmic powers of 2: $\pm$ [0.5, 1, 2, 4, 8, 16, 32, 64, 128, 256,..] m. (c, f) temperature differences: (c) reflective years from the 1979-2013 climatological mean (i.e., blue contours indicate regions where temperature is cooler than the climatology); (f) absorptive years from the climatological mean. The contour interval is 1 K and stipplings denote differences significant at the 95% confidence level.



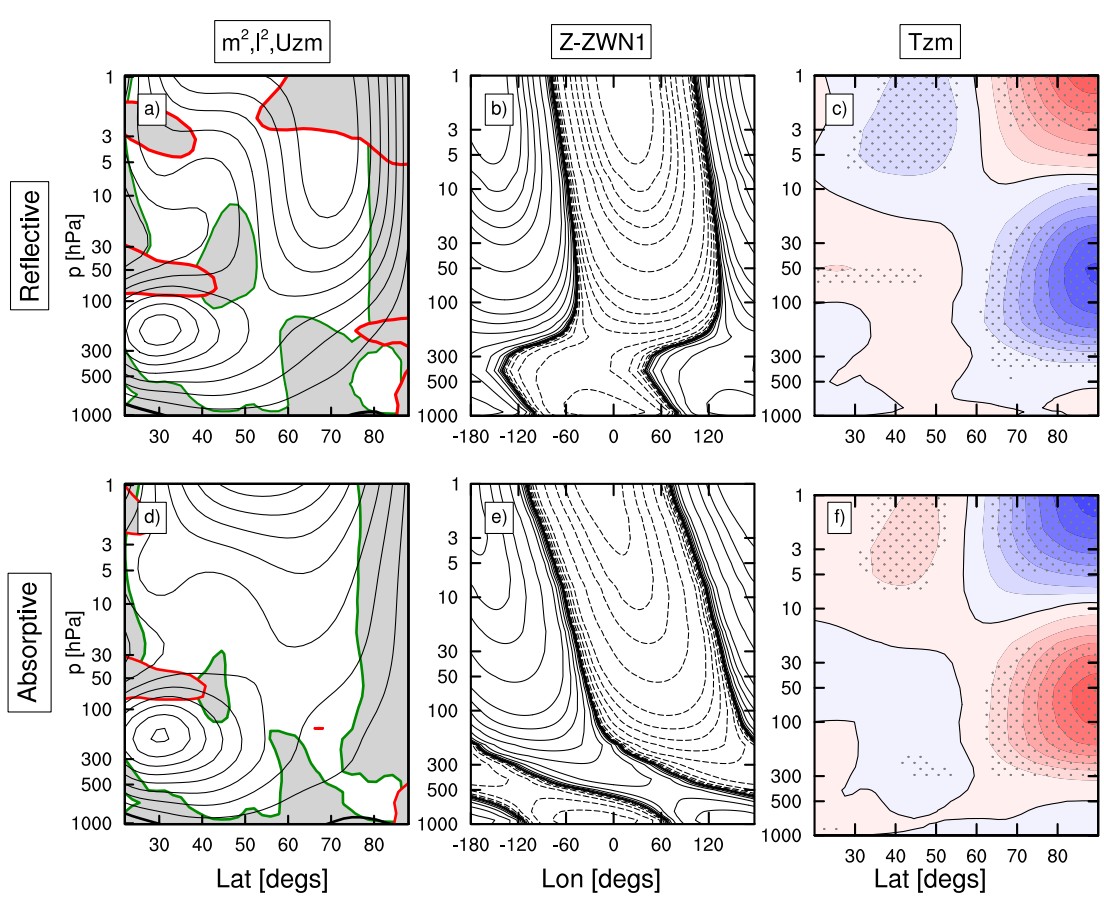

**Figure 11.** Same as Fig. 9, but from a 100-yr CESM1(WACCM) simulation.





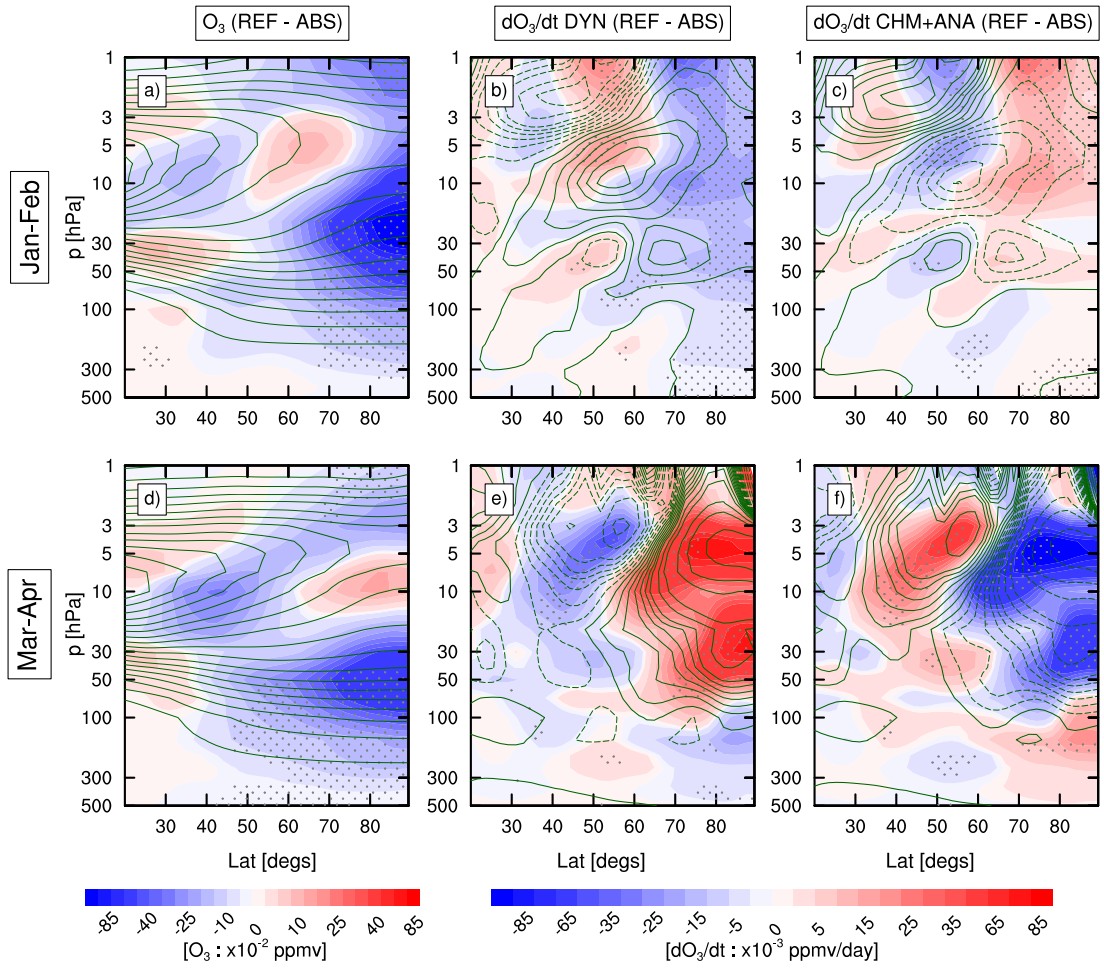

**Figure 12.** Differences in (a,d) zonal mean ozone, (b,e) ozone tendency due to dynamics, and (c,f) ozone tendency due to parameterized chemistry plus analysis between REF and ABS averaged between January-February (top) and March-April (bottom). The climatological mean values are denoted by green contour. The contour lines for this plot have increments of 5 units. The solid (dashed) lines indicate positive (negative values). The zero contour lines are plotted in thick darkgreen. Stipplings denote differences significant at the 95% confidence level.





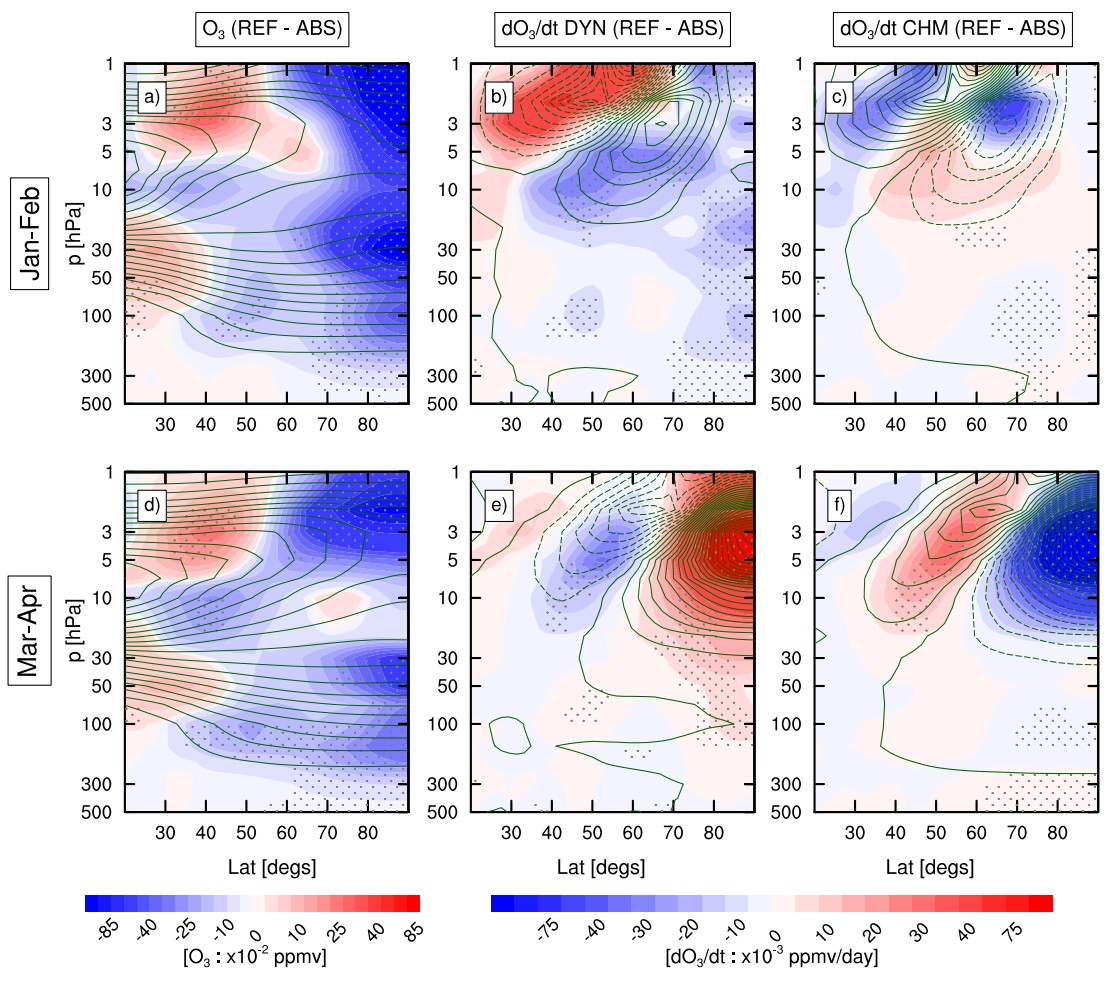

**Figure 13.** Same as Fig. 12, but from a 100-yr CESM1(WACCM) simulation.




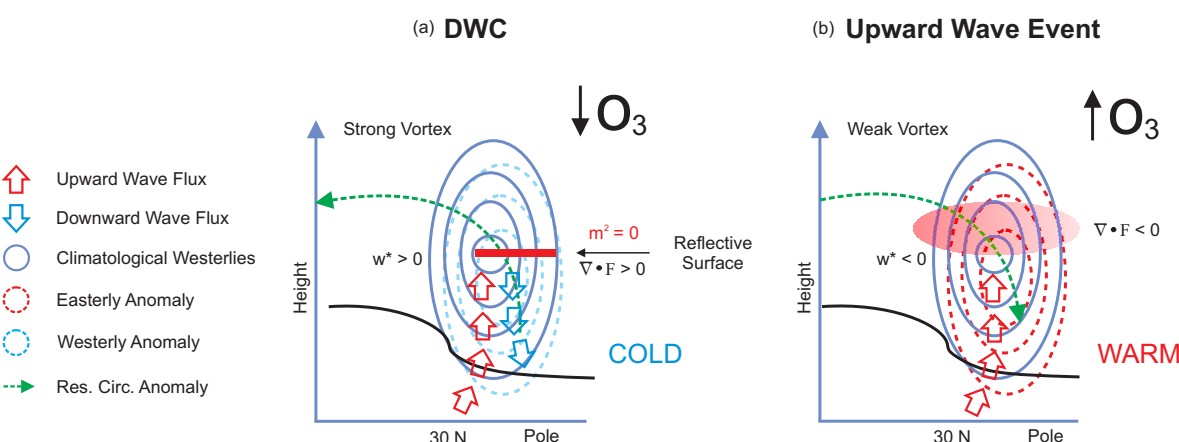

**Figure 14.** Schematic diagram depicting the mechanisms we propose that are responsible for changes in Arctic ozone levels in the winter associated with (a) DWC and (b) upward wave events. The red shading in Fig. 14b indicates EP flux convergence.





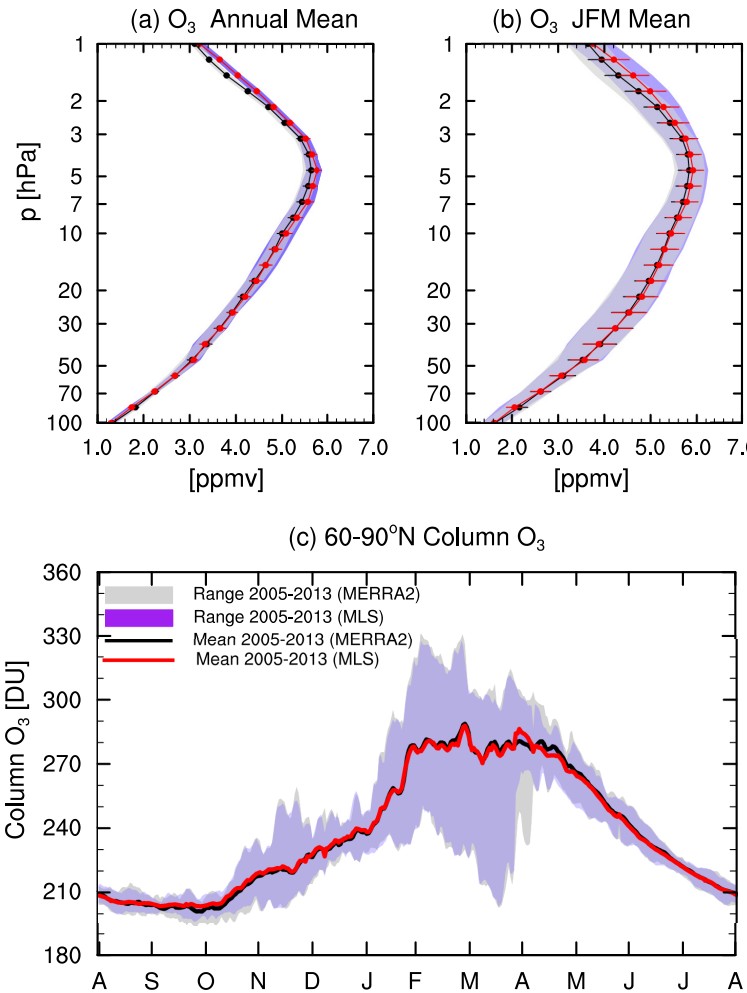

**Figure A1.** (a-b) Vertical profile of zonal mean ozone (ppmv) from MERRA2 and AURA MLS averaged between 60-90°N for (left) annual and (right) winter JFM means. (c) Daily climatology of total column Arctic ozone (vertically integrated from 100-1 hPa) in Dobson Units (DU). The solid black (red) curve denotes the climatological mean of MERRA2 (AURA MLS) ozone, and the gray (purple) shading indicates the range from MERRA2 (AURA-MLS) for the 2005-2013 time period.





**Figure A2.** Time series of total ozone (left column) and ozone anomaly (right column) averaged between 60-90°N as a function of height and time for (a,f) MERRA-1, (b,g) MERRA2, and (c,h) AURA MLS datasets from 2005 to 2013. Also plotted, the total ozone and ozone anomaly differences between MERRA-1 and AURA MLS (Fig. A2d and Fig. A2i, respectively), and the differences between MERRA2 and AURA MLS (Fig. A2e and Fig. A2i, respectively).