# Peer review of "How Does Downward Planetary Wave Coupling Affect Polar Stratospheric Ozone in the Arctic Winter Stratosphere?"

_Atmospheric Chemistry and Physics, 2016_

## Referee Comment (RC1) · Anonymous Referee #1 · 30 Aug 2016

**REVIEW: How Does Downward Planetary Wave Coupling Affect Polar Stratospheric Ozone in the Arctic Winter Stratosphere? by S. W. Lubis, V. Silverman, K. Matthes, N. Harnik, N-E. Omrani and S. Wahl**

In this manuscript the authors show that Downward Wave Coupling (DWC) events impact high-latitude stratospheric ozone in two ways: 1) reduced dynamical transport of ozone from low to high latitudes during individual events and 2) enhanced springtime chemical destruction of ozone via the cumulative impact of DWC events on polar stratospheric temperatures. The authors motivate the study by highlighting the focus

of previous work on the role of upward propagating waves. The results presented here broaden the scope of the impact of wave-mean flow interaction on stratospheric ozone by highlighting the key role of wave reflection. The authors make a convincing case supporting and extending previous published work.

My recommendation is that this manuscript should be published with minor revisions which are outlined in greater detail below.

**1   General comments:**

- I strongly recommend a reorganization of the figures in the main manuscript and the supplementary material. While it will not change the results of the paper I feel it would greatly improve the readability of the manuscript and highlight the key points in a simpler way. This would also reduce the total number of figures by 2. My recommendation is the following:

  – Figure 1: Show pressure-time plots of VT, EPFD, Psi, dtheta/dt for the DWC events where VT and EPFD panels are taken from the old fig. S1. This would remove the duplication of figures in fig. S1 and focus the reader on the dynamics of DWC events with a consistent set of axes.
  – Repeat the new fig.1 in fig. S1 for positive heat flux events.
  – Figure 2: combine the 2 line plots in the current fig.1 with the line plot in the current fig. 4. Again this would aid the reader with a consistent panel format in the figure.
  – Figure 3: Move the current fig. 2 to new fig. 3.
  – Figures 4: Don't change

   – Figures 5-8: Repeat the new organization from figs. 1-4 for the model.

- While the authors focus on wave-1 heat flux events, there is no mention of wave-2. How do the results in the first part of the paper compare to a similar analysis using wave-2 for both positive and negative heat flux events? Do reflective winters exhibit more extreme negative wave-2 events and vice versa for absorptive winters? Shaw and Perlwitz 2014 use the total eddy heat flux in their analysis of the role of DWC on stratospheric temperatures suggesting that wave-2 might also be important.

**2  Specific comments:**

- In terms of the definition of reflective winters, why not simply use the previously published definition from Perlwitz and Harnik 2003 based on the zonal-mean zonal wind shear? This index likely encapsulates both $U$ and $m^2$ criteria used here in a simple way. How does the PH03 shear index look when plotted next to the time series in figure 9?

- Are reflective winters dominated by DWC events? Similarly do "absorptive" winters contain a large amount of extreme positive heat flux events? Quantifying whether the individual events defined in section 1 occur in the seasons defined in section 2 would add additional support to the cumulative argument.

- I'm not sure I completely agree with the argument in the footnote of page 11 motivating the authors' removal of SSW's during "reflective" winters. It would be good if the authors could address the following points:

   – SSW's are effectively a continuum that depends on the definition threshold (e.g. Butler et al. 2015 BAMS fig. 4), and so the current definition includes

years that contain events that are very close to satisfying the SSW criteria, which would contribute to the season being "absorptive" when it is defined as "reflective".

– Doesn't the fact that seasons with SSW's can have increased wave reflection call into question the definition of "reflective" and "absorptive" winters based on the seasonal mean wind and wave geometry?

– If you include SSW years but keep the rest of the definition the same, do you get similar results? Perlwitz and Harnik 2003 did not exclude SSW years.

- L198-200: Is there a difference when time averaging from day -10 to +5 from figure 1b (i.e. all days not only significant days)?

- L343-344: How do you calculate m2 and l2? everyday and then average or from the average U and T? Does it make a difference? Are the results sensitive to the meridional/vertical averages and thresholds used in the definition?

- L364-365: Why is a wider meridional wave guide favorable for upward wave events? Presumably a narrower waveguide would focus wave activity polewards rather than equatorwards enhancing the positive heat fluxes. This seems to run counter to previous literature which argues that a strong poleward shifted vortex is conducive to upward wave activity (McIntyre 1982 and others).

- L412-413: What is meant by "sharpened gradients of ozone" and their relationship to dynamical terms? In addition, it seems that in the reflective winters during MA there are anomalous positive heat fluxes enhancing the climatological transport of ozone. This runs counter to the expectation during "reflective" winters.

- Figures 3 and 7: Why not simply plot all the data in the scatter plot instead of contours? The contours seem unnecessarily complicated and plotting the entire time series would allow the reader to see the large correlations from the full time series quoted in the text.

- Figure 14: I'm not sure the current schematic is helping to succinctly convey the dynamical mechanisms detailed in the text. I suggest either streamlining it or completely removing it since it is not very helpful in its current iteration.

**3   Technical corrections:**

L34: Suggest changing "it represents" to "proportional to"

L49: "increases" should be "increased"

L137: "transform" should be "transformed"

L138: Suggest putting the equation number from Andrews et al. 1987

L140: I suggest adding a sentence linking Equation 1 and 2, i.e. the first 3 RHS terms in equation 2 sum to the 2nd RHS term in equation 1. Same for chemistry and analysis terms.

L155: Suggest also citing Dunn-Sigouin and Shaw 2015 for the event definition

L159: Suggest citing Shaw et al. 2010 for period of maximum vertical wave coupling.

L22: Fig. "2e" should be "2f"

L237: "circular" should be "oval"

L270: You could also add the point that the historical time series is short and so you can get a more robust sample of events in the model to support the reanalysis dynamics.

L307-309: Perhaps mention that the detailed analysis of why the model terms are biased is beyond the scope of the article.

L321-322: Suggest mentioning the correlation for the entire data set in the model as was done for reanalysis data

L341: "ozones" should be "ozone"

L361-362: "due to enhanced DWC events": could also be due to enhanced equator-ward refraction and a lack of upward propagation

L398-399 and L409: Why "not shown"?
* * *

---

## Referee Comment (RC2) · Anonymous Referee #3 · 8 Sep 2016

This manuscript investigates the effect of downward planetary wave coupling (DWC) events on Northern Hemisphere polar stratospheric ozone in MERRA-2 reanalysis data and WACCM simulations. The authors analyze the DWC modulation of O3 via a direct effect through changes in the residual circulation and transport, and an indirect effect, through changes in polar temperature and chemistry; and show that the direct effect dominates in explaining the changes in O3 during DWC events. Finally, the authors analyze the seasonal impact of DWC events (reflective Winters).

I find this study interesting and adequate for publication in ACP after some minor revisions. In particular, reorganization of Figures, improvement of the comparison of model and reanalysis results, and better description of the results linking them to the direct

and indirect effects as discussed in the Introduction. Detailed comments are listed below.

L. 104 and L. 108. Please explain a bit more what is the correcting tendency term.

L. 113. The period is not so clear as in line 91 it says 1980-2013 and in line 94, it says 1978 to 2004. I understand it is 1980-2013 but it would be better to clarify.

L. 135. Please add a bit more detail on the simulation of volcanic eruptions in CESM1 (WACCM), see for instance Marsh et al. (2013).

L. 155. Define total . Is this the climatology plus anomaly?

L. 157. How the results compare with DJF? I wonder if for the upward events, the coupling is larger in DJF than in JFM.

L. 164. It would be good to add a sentence on the comparison of the frequency of events, which is actually pretty similar between the reanalysis and the model.

L. 170. Can you please explain briefly the Monte Carlo test? , so that the reader does not neccesarily need to check the references?

L. 187. I notice Fig.S1c and d are the same as Fig.1a and d. In addition, I consider the results on v*T* and the divergence of the EP flux are important enough to be in the main figures (not in the supplementary material). Please include those panels in Fig.1 and then remove Fig.S1 from the supplementary material.

L. 198. Which levels are the authors referring to? For each day the gray areas in Fig. 1a and 1b?

L. 211-212. I don't see this transition in Fig. 2 a. I see the change from positive (day -5) to negative (day 0) but both maximum and minimum are at the same altitude (around 10hPa), so I don't see the change from the upper stratosphere to the lower stratosphere.

L. 220-223. Again, here changes in ozone tendency due to dynamics are discussed between the midlower stratosphere and the mid-upper stratosphere. I don't see that .

L. 225-226.' . . . is evidente in the upper stratosphere'. This is actually only true in the days before the DWC.

L. 228. '. . .are relatively small. . .' They are actually not significant except for those around day -7. I think the description related to Fig. 2c needs to be improved.

L. 231 and L. 248 (and description of Figure 3c and 7c). L. 231 says that the same conclusion can be Dracn by assessing the instantaneous corrleation between for upward and downward heat flux events. I don't see this conclusion from the 3 panels in Fig. 3. I think it is obvious that whatever relationship between w* and O3 is going to be associated to the dynamical term in equation [2] and not with the chemistry effects (which are related to production and loss). Am I missing something? So I do not see the point in showing panel c in Figures 3 and 7. I would keep these figures with 2 panels each.

L. 259 and others. I don't fully understand what the authors mean by 'reversible or irreversible 'throughout the life cycle. Can you explain in hte manuscripts what are the consequences of having a reversible or irreversible impact? Reversible means that even though the impact is e.g. negative, it can become positive in the future? Please explain.

L. 264-265. I am not sure this sentence is right. I think it would be right if the time integration of ozone over the life cycle for DWC was negative , so it would balance the positive during upward events. But because Fig. 4 shows a time integration of ozone over the life cycle close to 0 for DWC and positive for upward events, I thikn what it means is maybe to 'minimize' or ' decrease' the increase in ozone, but not to 'prevent'.

L. 289. There are quite large differences in the values of Fig. 5c and 5d compared to 1c and 1d, with larger values in the model compared to MERRA. These differences are

not mentioned in the text. Please discuss them.

L. 325-334. Discussion of Fig. 8. I find interesting the differences in the evolution of total column ozone tendency the days previous to day 0 (significant blue lines in the figures), values about 1 DU/day in the model versus 5 in MERRA-2. Please discuss these differences and the possible reasons for them in the text.

L. 333. Again, I think 'prevent' is not the best Word here.

L. 343. How is m computed?

L. 345. Is sigma the sigma for JFM or which one?

L. 358. I see this tilt from 500hPa to about 100hPa but not up to the middle stratosphere.

L. 394 and discussion of Figure 12. I think it would help to add contour labels to the colors in the plots. It seems to me that the Dynamical and chemical terms cancel each other at the polar latitudes in Fig. 12b and c and 12e and f. Also the green contours mentioned in the caption seem black. Maybe better just to draw them in black.

Regarding Fig. 12, I wonder how symmetric or linear the response is between REF and ABS winters, otherwise it's hard to know if the negative signal in Fig. 12a comes from positive anomalies in ABS or negative anomalies in REF. Can the authors discuss how the individual signals (ABS and REF) are to make sure the description of the differences make sense?

L. 400, should it be 'in the upper polar stratosphere? Also, L. 401 talks about the signal at 10hPa but neither fig. 12a nor 12 d show significant signal at 10hPa. Please focus the description on the significant signals.

Section 5. Conclusions. I think the first 4 points could be combined. My understanding is that the direct effect described in the Introduction is the one related to tranport and w*, while the indirect effect is that associated with the chemistry and their dependency

on temperature. If this is correct, then it would make more sense to arrange the first four conlsusions putting together these results on w* on one hand , and the results on chemistry on the other. I miss the link between the direct and indirect effect discussed in the introdution with the actual results of the paper.

L. 445. Shouldn't be a positive divergence anomaly drawn in panel a of Figure 14, analogous to the negative anomaly in EP flux divergence in panel b?

L. 471. I think also a better understanding on stratospheric conditions , right? As it was shown here that the wave geometry in the stratosphere matters.

L. 481. Figures 1a and 1b should be Figures A1a and A1b.

L. 493. Figures A1c.

Figures: Please indicate in Fig captions 1 to 4 that those are with MERRA2 data.

Figure 9. Add in the caption which line is which (shaded or line plot).

Figure 10. What is the author referring with wave geometry in the caption of Figure 10?

---

## Referee Comment (RC3) · Anonymous Referee #4 · 15 Sep 2016

The manuscript analysis the instantaneous and cumulative impact of (Downward Wave Coupling) DWC events on Arctic ozone. While it is well known that planetary waves impact polar ozone through modulations of the residual circulation and stratospheric temperatures, the study adds new aspects by explicitly investigating the impact of DWC events on ozone via vertical advection and eddy transport. The analysis is convincing and highlights the role of wave reflection for high latitude ozone field. I recommend publication after addressing the following comments.

1) Directly comparing the impact of DWC events on ozone with the impact of upward wave events somehow implies that positive wave activity (as seen before the DWC) without the occurrence of wave reflection would turn into such strong upward wave

events. The line of argumentation (and in particular the wording '... prevents ...') re-inforces this impression (e.g., line 265). Please clarify if DWC events indeed prevent large positive ozone anomalies as shown in Figure 4 and 8. If this cannot be clearly concluded revise the text accordingly.

2) Some aspects of Figures 2 and 6 are not discussed in the text. Why does the vertical advection imply negative ozone tendencies in the upper stratosphere in case of upward wave events? This can be discussed by using the shape of the ozone VMR profile. What about the instantaneous chemical response in the upper stratosphere? Why does the upper stratospheric signal of the eddy transport from upward wave events last much longer in the model simulation?

3) It is not at all clear why at the end of the winter the dynamical composite gives positive ozone anomalies for reflective winters (Figure 12e). This needs to be clearly explained and related to the weaker or reversed ozone transport. The argument from line 413 is confusing since sharpened meridional gradients should even more so result in negative anomalies in case of less transport.

4) The positive chemical ozone anomaly in reflective year's midwinter (Figure 12c) is not mentioned or explained in the text.

5) Please explain if Figure 1 is based on a DWC composite or a single DWC event?

6) Line 190: Deceleration doesn't necessarily result in adiabatic cooling.

7) Line 211-212: The negative and positive values seemed to be almost at the same level (instead of in the 'upper' and 'lower' stratosphere.)

8) Line 216: Or transport out of the polar vortex?

9) How were the terms in Figure 2 calculated? Are the DYN and CHM+ANA composites MERRA-2 output? Are the composites in 2d-2f calculated with equation 2? Are the terms consistent (i.e. is the sum of 2d-2f the same as DYN)?

10) Line 395: Is the direct effect mentioned here the effect over the DWC life cycle? This was shown to be nearly zero and not a 'weak increase'.

Minor comments:

1) Line 95: AURA -> Aura

2) Line 169: Please explain the minimum ozone term in Table 1. Is this the ozone tendency related to the date of the DWC (and this not necessarily a temporal minimum)?

3) Line 222: 2e -> 2d

4) Caption Figure 1: a-d -> a-b

---

## Author Comment (AC1) · 12 Dec 2016

**S.W. Lubis1, V. Silverman3, K. Matthes1, 2, N. Harnik3, N. Omrani1, 2, S. Wahl1**

Correspondence to: S.W. Lubis (slubis@geomar.de)

**1. Reviewer #1**

In this manuscript the authors show that Downward Wave Coupling (DWC) events impact high-latitude stratospheric ozone in two ways: 1) reduced dynamical transport of ozone from low to high latitudes during individual events and 2) enhanced springtime chemical destruction of ozone via the cumulative impact of DWC events on polar stratospheric temperatures. The authors motivate the study by highlighting the focus of previous work on the role of upward propagating waves. The results presented here broaden the scope of the impact of wave-mean flow interaction on stratospheric ozone by highlighting the key role of wave reflection. The authors make a convincing case supporting and extending previous published work. My recommendation is that this manuscript should be published with minor revisions, which are outlined in greater detail below.

We thank the reviewer for reading the manuscript and providing their helpful comments and suggestions. We address their specific issues in turn below.

**1.1 General comments**

1. I strongly recommend a reorganization of the figures in the main manuscript and the supplementary material. While it will not change the results of the paper I feel it would greatly improve the readability of the manuscript and highlight the key points in a simpler way. This would also reduce the total number of figures by 2. My recommendation is the following:

- Figure 1: Show pressure-time plots of VT, EPFD, Psi, dtheta/dt for the DWC events where VT and EPFD panels are taken from the old fig. S1. This would remove the duplication of figures in fig. S1 and focus the reader on the dynamics of DWC events with a consistent set of axes.
- Repeat the new fig.1 in fig. S1 for positive heat flux events.
- Figure 2: combine the 2 line plots in the current fig. 1 with the line plot in the current fig. 4. Again this would aid the reader with a consistent panel format in the figure.
- Figure 3: Move the current fig. 2 to new fig. 3.
- Figures 4: Don't change
- Figures 5-8: Repeat the new organization from figs. 1-4 for the model.

We agree with the reviewer's suggestion. We have now modified Fig.1 by adding the evolution of V'T' and EPFD from the old Fig. S1 to this figure (see Fig. 1). We have also now combined the 2 line plots in the old Fig.1 with the line plot in the old Fig. 4 (see Fig. 4). We repeated the new organization from Fig.1-4 for the model (see Fig. 5-8). In addition, we have now included the evolution of V'T', EPFD, Psi, and dtheta/dt, ozone time-tendency terms for upward wave events from both MERRA2 and CESM1 (WACCM) in the supplemental material (see Fig. S1 to Fig. S4). Correspondingly, we changed the order of the

<sup>1GEOMAR Helmholtz Centre for Ocean Research Kiel, Kiel, Germany

<sup>2Christian-Albrechts Universität zu Kiel, Kiel, Germany

<sup>3 Geophysical Institute, University of Bergen and Bjerknes Centre for Climate Research, Bergen, Norway

<sup>4 Department of Geophysical, Atmospheric and Planetary Sciences, Tel Aviv University, Israel

presentation to better fit the order of appearance in the figures (combined the discussion of reversibility of the overturning circulation, temperature and ozone to after figures 1-3). Furthermore, as suggested by reviewer #3, for a better comparison between the reanalysis and model, we have now combined the old Fig. 11 into old Fig. 10 (see new Fig. 10), and the old Fig. 13 into old Fig. 12 (see new Fig. 11). After doing these changes, the total number of figures is reduced by 2, as expected by reviewer #1.

**2. While the authors focus on wave-1 heat flux events, there is no mention of wave-2. How do the results in the first part of the paper compare to a similar analysis using wave-2 for both positive and negative heat flux events? Do reflective winters exhibit more extreme negative wave-2 events and vice versa for absorptive winters? Shaw and Perlwitz 2014 use the total eddy heat flux in their analysis of the role of DWC on stratospheric temperatures suggesting that wave-2 might also be important.**

We focus on wave-1 because it dominates the total eddy (deviation from zonal-mean) heat flux in the stratosphere and it represents the dominant source of downward wave coupling between the stratosphere and troposphere (e.g., Perlwitz and Harnik 2003, Shaw et al, 2010, Lubis et al., 2016a). Nevertheless, we also examined the impact of downward wave-2 coupling on Arctic ozone (see Fig S5 and Fig. S6 in supplement). The results showed that the stratospheric impacts associated with downward wave-2 events on ozone are not as robust as the corresponding downward wave-1 events (see Fig. S6). A possible reason for this is the evolution of the wave-2 events involves weaker heat flux values and weaker EPFD (Fig. S5, consistent with Dunn-Sigouin and Shaw, 2015), which results in a weaker connection to the total ozone time tendency, due to weaker changes in the associated residual circulation and temperature tendency.

We have included the discussion regarding the relation of wave-2 and ozone in the text (see **P8 L32**) and provided the supporting figures in supplemental material (see Figs. S5 and S6).

**1.2 Specific comments**

**1.** In terms of the definition of reflective winters, why not simply use the previously published definition from Perlwitz and Harnik 2003 based on the zonal-mean zonal wind shear? This index likely encapsulates both U and m2 criteria used here in a simple way. How does the PH03 shear index look when plotted next to the time series in figure 9?

We actually checked both reflective index definitions in the original version of the manuscript. We agree that zonal wind shear index (based on U2-10) can be used to determine the reflecting surface (similar as  $m^2$ ), but that alone cannot directly determine the strength of the polar vortex.

REF is defined as winters characterized by (1) a formation of reflecting surface above the mid-stratospheric jet (i.e., polar vortex is strong, see Fig. 11a of Perlwitz and Harnik 2003, hereafter refer to PH03) and (2) dominated by downward wave activity (i.e., less wave absorption). To encapsulate these two criteria, the m2 is used to track the location of turning (reflecting) surface, while the mid-stratospheric U30 is used to measure the strength of the polar vortex. This method is similar as that used by Harnik (2009) to defined REF, where they included additional criteria to the U2-10 index in order to ensure that the vortex in REF remains strong.

The problem of using the U2-10 index alone is that negative U2-10 values sometimes occur during or after sudden warming events (see also our discussion in the comments below).

Thus, in defining a reflection index based on U2-10 (PH03), care should be taken to only look at negative U2-10 events for which winds in the lower stratosphere remain **strong** (Harnik 2009). As in Harnik's 2009 study, the winters dominated by SSW events are categorized as absorptive winters (ABS), due to the fact that the polar vortex is weaker (i.e., indicated by weaker U30 values) and is dominated by strong wave absorption in the stratosphere. We also note that the number of wave reflection events during/after SSW events is less compared to the amount of wave absorption in the stratosphere. Therefore, it is worth to use both m2 as an indicator for vertical reflecting surface and U30 as an indicator for vortex strength, in order to define the REF in our study.

To support our argument, these two types of definition are compared here. **Figure R1** (below) shows the comparison between REF/ABS based on our definition, and those based on PH03. In general, most of ABS and REF years defined in our study are consistent with PH03. However, some discrepancies exist, in particular for the REF winters: (1) 1997 and 2011 are grouped as REF in our study, but are not captured by PH03 because the reflecting surface is slightly higher (i.e., it is not well captured by U2-10 index). However, when using the U2-10 index, these winters can be grouped as REF, and (2) 1992, 2000, and 2003 are not categorized as REF in our study, because major warmings occur during this winter, as indicated by a weaker polar vortex (i.e., dominant wave absorption).

**Fig. R1**. Comparison of REF and ABS definitions based on our definition (top) and based on PH03 (bottom). The dark green horizontal lines indicate the region of the long-term average +/-0.25 std deviation.

2. Are reflective winters dominated by DWC events? Similarly do "absorptive" winters contain a large amount of extreme positive heat flux events? Quantifying whether the individual events defined in section 1 occur in the seasons defined in section 2 would add additional support to the cumulative argument.

Yes, the reflective (absorptive) winters are dominated by DWC (upward wave) events characterized by a large amount of extreme negative (positive) heat flux events during this period (see Fig. S7 in supplement). We have now included this figure in the supplementary material (see Fig. S7) and explained this in the text (see P11, L30).

3. I'm not sure I completely agree with the argument in the footnote of page 11 motivating the authors' removal of SSW's during "reflective" winters. It would be good if the authors could address the following points:

- SSW's are effectively a continuum that depends on the definition threshold (e.g. Butler et al. 2015 BAMS fig. 4), and so the current definition includes years that contain events that are very close to satisfying the SSW criteria, which would contribute to the season being "absorptive" when it is defined as "reflective".
- Doesn't the fact that seasons with SSW's can have increased wave reflection call into question the definition of "reflective" and "absorptive" winters based on the seasonal mean wind and wave geometry?

We agree with the reviewer that there is a continuum of events, in the sense that wave reflection events start with a deceleration in the upper stratosphere, whereas in sudden warmings the deceleration extends lower down, and that the level to which such deceleration reaches can occupy a continuum. A very relevant recent study by Kodera et al. (2016) showed that a subset of SSW events end with downward wave reflection. Comparing reflective and non-reflective SSWs they showed that while reflective SSWs are characterized by a quick termination of the warming episode due to the reflection of planetary waves, the SSWs events for which waves are not reflected have a longer timescale. The reflective SSW events can be viewed as a mixture between SSW and reflection events, as pointed out by the reviewer. We think however that these events may involve over-reflection rather than reflection, because a critical surface forms embedded by two opposite signs of PV gradient, thus they should be studied separately. In particular, it remains to be seen if the effect of the SSW event on ozone levels is reversible as it is in REF events or not (we expect it to be only partially reversible). This however is beyond the scope of this study, and to isolate the indirect effect of shorter/weaker wave reflection events on ozone via an influence on spring PSC concentration, we chose to exclude these reflective SSW events because of their different effect on polar temperatures. We have now removed the footnote and clarify the definition of REF winter with/without the inclusion of SSWs events (see P11 L26).

**4. If you include SSW years but keep the rest of the definition the same, do you get similar results? Perlwitz and Harnik 2003 did not exclude SSW years.**

There are no large differences *in the patterns* of wave geometry and temperature between the composites of reflective years which include and do not include SSW (see Fig. R2 bellow). In particular, it is shown that the structures of wave geometry and temperature in REF and ABS based on PH03 are in agreement with those composited using our criteria, characterized by elongated vertical reflecting surface during REF and termination of reflecting surface in ABS (the reflecting surface shifts poleward away from the meridional waveguide, Fig. R2). However, the strength of the temperature response in the polar lower to mid stratosphere based on PH03 index is somewhat weaker and not as robust, due to the inclusion of SSWs. Therefore, it is important to define REF winter where the vertical wind shear is negative, BUT the winds in the mid-lower stratosphere remain **strong**. As we discussed above, we used an extended definition of PH03 as in Harnik (2009), by separating the reflective events from the absorptive events associated with SSWs. We have now clarified and emphasized this in our manuscript (see P11, L25-32).

---

## Author Comment (AC2) · 12 Dec 2016

**S.W. Lubis[1], V. Silverman[3], K. Matthes[1, 2], N. Harnik[3], N. Omrani[1, 2], S. Wahl[1]**

[1] GEOMAR Helmholtz Centre for Ocean Research Kiel, Kiel, Germany
[2] Christian-Albrechts Universität zu Kiel, Kiel, Germany
[3] Geophysical Institute, University of Bergen and Bjerknes Centre for Climate Research, Bergen, Norway
[4] Department of Geophysical, Atmospheric and Planetary Sciences, Tel Aviv University, Israel

Correspondence to: S.W. Lubis (slubis@geomar.de)

**2. Reviewer #3**

*This manuscript investigates the effect of downward planetary wave coupling (DWC) events on Northern Hemisphere polar stratospheric ozone in MERRA-2 reanalysis data and WACCM simulations. The authors analyze the DWC modulation of O3 via a direct effect through changes in the residual circulation and transport, and an indirect effect, through changes in polar temperature and chemistry; and show that the direct effect dominates in explaining the changes in O3 during DWC events. Finally, the authors analyze the seasonal impact of DWC events (reflective Winters). I find this study interesting and adequate for publication in ACP after some minor revisions. In particular, reorganization of Figures, improvement of the comparison of model and reanalysis results, and better description of the results linking them to the direct and indirect effects as discussed in the Introduction. Detailed comments are listed below.*

We thank the reviewer #3 for her/his constructive comments and very close reading of our manuscript. We have made substantial modifications that we hope have clarified our paper.

**2.1 Specific comments**

*1. L. 104 and L. 108. Please explain a bit more what is the correcting tendency term.*

The analysis term (i.e., correcting tendency term) is part of the Incremental Analysis Update (IAU) (Bloom et al, 1966), which is used in the GEOS5 model and is an additional forcing to constrain the model to the observations. We added this information in the text (**see P3. L31**).

*2. L. 113. The period is not so clear as in line 91 it says 1980-2013 and in line 94, it says 1978 to 2004. I understand it is 1980-2013 but it would be better to clarify.*

MERRA-2 assimilates satellite observations from the SBUV from 1980 to 2004, and from October 2004 from OMI and MLS (Bosilovich 2015). We have clarified this in the text (**see P4. L8**).

*3. L. 135. Please add a bit more detail on the simulation of volcanic eruptions in CESM1 (WACCM), see for instance Marsh et al. (2013).*

Observed volcanic eruptions of the twentieth century are included by prescribing a monthly zonal-mean time series of volcanic aerosol surface area density (SAD), identical to that used in the CCMVal2 REF-B1 simulations (**see P4. L25**).

*4. L. 155. Define total. Is this the climatology plus anomaly?*

Yes, the total is defined as the climatology plus anomaly (**see P5, L16**).

**5. L. 157. How the results compare with DJF? I wonder if for the upward events, the coupling is larger in DJF than in JFM.**

We focus our analysis during JFM because it represents the dominant time period of maximum downward planetary wave coupling in the Northern Hemisphere (Shaw et al., 2010, Lubis et al., 2013). Qualitatively similar results are found using the extended DJF and NDJFM winter season (consistent with **Dunn-Sigouin and Shaw 2015**).

**6. L. 164. It would be good to add a sentence on the comparison of the frequency of events, which is actually pretty similar between the reanalysis and the model.**

We have now explicitly mentioned it in our manuscript. The frequency of DWC events in MERRA and CESM is similar, about 6 events per decade. (**see P5. L24**).

**7. L. 170. Can you please explain briefly the Monte Carlo test? , so that the reader does not neccesarily need to check the references?**

We have now included this in Appendix C.

**8. L. 187. I notice Fig.S1c and d are the same as Fig.1a and d. In addition, I consider the results on v\*T\* and the divergence of the EP flux are important enough to be in the main figures (not in the supplementary material). Please include those panels in Fig.1 and then remove Fig.S1 from the supplementary material.**

We have now modified the figure and the text accordingly (see our general comments in the first page).

**9. L. 198. Which levels are the authors referring to? For each day the gray areas in Fig. 1a and 1b?**

The levels refer to the region where the signals are statistically significant (i.e., between 100-1 hPa, gray areas in Fig. 1a and 1b). We have now clarified this in the text (**see P7. L9**).

**10. L. 211-212. I don't see this transition in Fig. 2 a. I see the change from positive (day -5) to negative (day 0) but both maximum and minimum are at the same altitude (around 10hPa), so I don't see the change from the upper stratosphere to the lower stratosphere.**

We apologize for this oversight. We agree that the positive ozone tendency subsequently changes sign and reaches its minimum value at 10 hPa, around day 0. We have revised the text accordingly (**see P7. L13**).

**11. L. 220-223. Again, here changes in ozone tendency due to dynamics are discussed between the mid-lower stratosphere and the mid-upper stratosphere. I don't see that.**

For clarity, we have included the pressure levels into the text (**see P7. L8-9**).

**12. L. 225-226.' . . . is evident in the upper stratosphere'. This is actually only true in the days before the DWC.**

We apologize for this oversight. We agree that the contribution of the chemistry to the total ozone tendency is evident in the upper stratosphere before the mature stage of DWC, from days -10 to -5. We have revised the text accordingly (**see P7. L14**).

**13. L. 228. '. . .are relatively small. . .' They are actually not significant except for those around day -7. I think the description related to Fig. 2c needs to be improved.**

We have revised the description related to Fig. 2c (**see P7. L14**).

**14. L. 231 and L. 248 (and description of Figure 3c and 7c). L. 231 says that the same conclusion can be drawn by assessing the instantaneous correlation between for upward and downward heat flux events. I don't see this conclusion from the 3 panels in Fig. 3. I think it is obvious that whatever relationship between w\* and O3 is going to be associated to the dynamical term in equation [2] and not with the chemistry effects (which are related to production and loss). Am I missing something? So I do not see the point in showing panel c in Figures 3 and 7. I would keep these figures with 2 panels each.**

We have now clarified this in our text that the instantaneous link between ozone and extreme wave-1 heat flux events is more dominated by the dynamical process, consistent with the results from Fig.2 where the transient changes of ozone during the life cycle of DWC is mainly due to changes in ozone transport (**see P8, L4**).

**15. L. 259 and others. I don't fully understand what the authors mean by 'reversible or irreversible' throughout the life cycle. Can you explain in the manuscripts what are the consequences of having a reversible or irreversible impact? Reversible means that even though the impact is e.g. negative, it can become positive in the future? Please explain.**

A wave packet passing through a medium will induce EP flux convergence at its head and EP flux divergence at its tail, and the time integrated EP flux divergence will be zero (a reversible effect of the waves on the mean flow), assuming there is no dissipation or no nonlinearities (the non-interaction theorem). Thus, we expect the effect of a wave which propagates to the stratosphere and then gets reflected back down will be more reversible than that of a wave that gets absorbed in the stratosphere via nonlinear wave breaking and a cascade to small scales which get dissipated. With this in mind, reversible means that the effect of DWC on ozone is canceled out over the life cycle of the wave, as indicated by the time tendencies of ozone that change from being positive to negative. Thus the overall effect of having more DWC events in winter is to have lower ozone levels in the polar stratosphere (i.e., DWC weakens the typical increase of ozone induced by upward wave propagation). On the other hand, the effect of upward wave event on ozone is irreversible over the life cycle, with the time tendencies not reversing during the life cycle. This means that increased upward wave events result in increased ozone concentration in the polar stratosphere due to stronger transport. We have clarified this in our text (**see P8. L14, P8. L25).**

**16. L. 264-265. I am not sure this sentence is right. I think it would be right if the time integration of ozone over the life cycle for DWC was negative, so it would balance the positive during upward events. But because Fig. 4 shows a time integration of ozone over the life cycle close to 0 for DWC and positive for upward events, I think what it means is maybe to 'minimize' or ' decrease' the increase in ozone, but not to 'prevent'.**

As stated in the manuscript, the impact of DWC events on ozone is transient and involves *a positive to negative total ozone tendency evolution*, where the total net effect (as shown by time integration) is nearly zero. This means the impact of DWC on ozone is reversible. Therefore, increase DWC events in winter weaken the typical increase of ozone induced by upward wave events. We have now used the correct word for this in our text (**see P8. L29**).

**17. L. 289. There are quite large differences in the values of Fig. 5c and 5d compared to 1c and 1d, with larger values in the model compared to MERRA. These differences are not mentioned in the text. Please discuss them.**

We agree with the reviewer that there are significant differences in the values of the temperature tendency between model and reanalysis. However, the differences in the residual circulation anomaly between model and reanalysis are relatively small. The reason for the discrepancy in temperature is due mainly to bias in modeled temperature in WACCM. In particular, WACCM still exhibits a bias in the stratospheric westerly jets and polar temperatures in the NH winter, where the largest biases in the stratosphere are in the location of the maximum of the NH westerly jet (Marsh et al., 2013). This bias, however, could be reduced by increasing non-orographic gravity wave drag, but at the cost of a less realistic mesopause (see Richter et al., 2010 and Marsh et al., 2013). We have discussed this in the text (**see P10. L28**).

*18. L. 325-334. Discussion of Fig. 8. I find interesting the differences in the evolution of total column ozone tendency the days previous to day 0 (significant blue lines in the figures), values about 1 DU/day in the model versus 5 in MERRA-2. Please discuss these differences and the possible reasons for them in the text.*

We agree with the reviewer that there are differences in the time evolution of ozone tendency prior to mature stage (day 0) of DWC between model and MERRA2. The positive ozone tendency values prior to DWC event persist longer in the model compared to MERRA2, which is consistent with the persistent poleward residual circulation anomalies. We have now discussed this in the text (**see P11. L5**).

*19. L. 333. Again, I think 'prevent' is not the best Word here.*

We agree with the reviewer. We have modified this sentence and others by replacing "to prevent" with "to weaken" (**see P11. L10**).

*20. L. 343. How is m computed?*

We have clarified this in the text (**see P11. L21**)

*21. L. 345. Is sigma the sigma for JFM or which one?*

The classifications are based on the vertical wave numbers (m) and zonal-mean zonal wind at 30 hPa (U30) in winter months (JFM). We have clarified this in the text (see. **P11. L25**).

*22. L. 358. I see this tilt from 500hPa to about 100hPa but not up to the middle stratosphere.*

We apologize for this oversight. Indeed, the eastward phase tilt with heights of the wave-1 structure is visible from the mid-troposphere to the lower stratosphere (**see P12. L10**).

*23. L. 394 and discussion of Figure 12. I think it would help to add contour labels to the colors in the plots. It seems to me that the Dynamical and chemical terms cancel each other at the polar latitudes in Fig. 12b and c and 12e and f. Also the green contours mentioned in the caption seem black. Maybe better just to draw them in black.*

We have now modified this figure by adding the contour labels, changing the intervals of the contour line and shading, and using another color table with better gradation (**see Fig. 11**). For a better comparison with the model, we have now combined this figure from MERRA2

with the figure from the model simulation. We do see that the CHM (DYN) terms are dominated the ozone tendency in REF during mid winter (late winter).

**24. Regarding Fig. 12, I wonder how symmetric or linear the response is between REF and ABS winters, otherwise it's hard to know if the negative signal in Fig. 12a comes from positive anomalies in ABS or negative anomalies in REF. Can the authors discuss how the individual signals (ABS and REF) are to make sure the description of the differences make sense?**

Yes the responses are symmetric. We have now discussed this in the text (**see P13 L1, P13 L12 and P13 L20**). The responses of ozone and ozone tendency in REF and ABS winters are symmetric with respect to climatological mean, so that negative ozone anomalies during mid winter or early spring indeed come from negative anomalies in REF.

**25. L. 400, should it be 'in the upper polar stratosphere? Also, L. 401 talks about the signal at 10hPa but neither fig. 12a nor 12 d show significant signal at 10hPa. Please focus the description on the significant signals.**

We have modified these sentences by focusing the interpretation of the results on the significant regions only (see **P13. L15**).

**26. Section 5. Conclusions. I think the first 4 points could be combined. My understanding is that the direct effect described in the Introduction is the one related to transport and w\*, while the indirect effect is that associated with the chemistry and their dependency on temperature. If this is correct, then it would make more sense to arrange the first four conclusions putting together these results on w\* on one hand, and the results on chemistry on the other. I miss the link between the direct and indirect effect discussed in the introduction with the actual results of the paper.**

We have modified our conclusion by focusing it into 4 key results: **(1)** The impact of DWC on the residual circulation and the temperature over the wave life cycle. This is important starting point to elucidate the direct and indirect impact of DWC on ozone. **(2)** The direct impact of DWC on ozone through residual circulation (*w\* or PSI*). **(3)** The indirect impact of DWC on ozone through the temperature changes (*dT/dt*). Finally, we close the conclusion by stating the cumulative (seasonal) impact of DWC on ozone in mid winter and early spring. We removed the point 2 in the old manuscript, since it is already included in the current point 2.

We think that our current conclusion has encapsulated the important key results found in our study.

**27. L. 445. Shouldn't be a positive divergence anomaly drawn in panel a of Figure 14, analogous to the negative anomaly in EP flux divergence in panel b?**

Yes, we have added a blue shading in panel (a), indicating a positive divergence anomaly (see **Fig. 12**).

**28. L. 471. I think also a better understanding on stratospheric conditions, right? As it was shown here that the wave geometry in the stratosphere matters.**

We have clarified this in the text (**see P15. L18**).

**29. L. 481. Figures 1a and 1b should be Figures A1a and A1b.**

Corrected.

***30. L. 493. Figures A1c.***

Corrected.

***31. Figures: Please indicate in Fig captions 1 to 4 that those are with MERRA2 data.***

We have now added this information in Fig captions 1 to 4.

***32. Figure 9. Add in the caption which line is which (shaded or line plot).***

We have now modified the caption. The shaded (line) indicates the *m* (*U30*).

***33. Figure 10. What is the author referring with wave geometry in the caption of Figure 10?***

We have now clarified the caption. The wave geometry configuration is referred to stratospheric configuration where a vertical reflecting surface (*m*) bounded above by a well-defined high-latitude waveguide (*l*).

===

---

## Author Comment (AC3) · 12 Dec 2016

**S.W. Lubis[1], V. Silverman[3], K. Matthes[1, 2], N. Harnik[3], N. Omrani[1, 2], S. Wahl[1]**

[1] GEOMAR Helmholtz Centre for Ocean Research Kiel, Kiel, Germany
[2] Christian-Albrechts Universität zu Kiel, Kiel, Germany
[3] Geophysical Institute, University of Bergen and Bjerknes Centre for Climate Research, Bergen, Norway
[4] Department of Geophysical, Atmospheric and Planetary Sciences, Tel Aviv University, Israel

Correspondence to: S.W. Lubis (slubis@geomar.de)

**3. Reviewer #4**

*The manuscript analysis the instantaneous and cumulative impact of (Downward Wave Coupling) DWC events on Arctic ozone. While it is well known that planetary waves impact polar ozone through modulations of the residual circulation and stratospheric temperatures, the study adds new aspects by explicitly investigating the impact of DWC events on ozone via vertical advection and eddy transport. The analysis is convincing and highlights the role of wave reflection for high latitude ozone field. I recommend publication after addressing the following comments.*

We thank the reviewer for reading the manuscript and providing their helpful comments and suggestions. We address their specific comments in turn below.

**3.1 Specific Comments:**

*1) Directly comparing the impact of DWC events on ozone with the impact of upward wave events somehow implies that positive wave activity (as seen before the DWC) without the occurrence of wave reflection would turn into such strong upward wave events. The line of argumentation (and in particular the wording '. . . prevents ...') reinforces this impression (e.g., line 265). Please clarify if DWC events indeed prevent large positive ozone anomalies as shown in Figure 4 and 8. If this cannot be clearly concluded revise the text accordingly.*

It is expected that increased wave absorption in the stratosphere (without an occurrence of wave reflection) result in stronger wave convergence (**Figs. S1b, S3b**) and thus, leads to a stronger transport of ozone to the polar region (i.e., increased ozone concentration). Our results (**Fig. 4 and Fig. 8**) confirmed this by showing that in the absence of DWC event (**red lines**), the ozone tendency in the polar stratosphere increases. This means that winters dominated by DWC events are characterized by lower ozone levels compared to those during absorptive winters. Nevertheless, we agree with the reviewer (as with reviewer #3), that the impact of DWC on ozone is actually to 'minimize/ weaken' the typical increase in ozone, but not to 'prevent' ozone increase due to upward wave events. We have changed our text thoroughly.

*2) Some aspects of Figures 2 and 6 are not discussed in the text. Why does the vertical advection imply negative ozone tendencies in the upper stratosphere in case of upward wave events? This can be discussed by using the shape of the ozone VMR profile. What about the instantaneous chemical response in the upper stratosphere? Why does the upper stratospheric signal of the eddy transport from upward wave events last much longer in the model simulation?*

As stated in the title, the paper is focused on the impact of DWC on polar stratospheric ozone. Therefore, the analysis of the upward wave event in this paper can be seen as a

complementary to DWC. Nevertheless, we appreciate the reviewer's suggestions and we have modified the text accordingly. Our specific responses are given below:

- *Why does the vertical advection imply negative ozone tendencies in the upper stratosphere in case of upward wave events?* By decomposing the vertical advection term into -w* and dO3/dz using a TEM continuity equation, we found that this negative anomaly is associated with negative vertical gradient of ozone (dO3/dz) at the upper stratosphere. This negative vertical gradient is expected, more ozone transported downward to the mid-lower stratosphere, leading to less ozone concentration at the upper stratosphere. The opposite behavior occurs during DWC events.

- *What about the instantaneous chemical response in the upper stratosphere?* The negative ozone tendency due to chemistry in the upper stratosphere prior to DWC is likely associated with increased chemical ozone loss due to increased temperature prior to DWC event. This is consistent with the ozone sink reaction in the upper stratosphere, which is more strongly dependent on temperature, where the increase (decrease) in temperature leads to more (less) ozone destruction (Brasseur and Solomon 2005). We have discussed this in the text (**see P7, L16**).

- *Why does the upper stratospheric signal of the eddy transport from upward wave events last much longer in the model simulation?* This associated with biases in the model in reproducing the evaluation of upward wave pulses prior to DWC. In the model, the upward heat flux is more persistent and lasts longer compared to the MERRA2. Since the eddy transport term (**M**) is dependent on the divergence of eddy heat flux term, the longer evolution of the eddy heat flux prior to DWC can lead to more persistent ozone tendency due to eddy transport. We have now discussed this discrepancy in association with a persistent heat flux signals in the model prior to DWC events (**see P9. L17 and P9. L32**).

*3) It is not at all clear why at the end of the winter the dynamical composite gives positive ozone anomalies for reflective winters (Figure 12e). This needs to be clearly explained and related to the weaker or reversed ozone transport. The argument from line 413 is confusing since sharpened meridional gradients should even more so result in negative anomalies in case of less transport.*

We have revised this particular part. Our results showed that the contribution of dynamics on ozone tendency in REF during early spring (late winter) is higher compared to ABS (see **Figs. S8k,l and Figs. S9k,l**).

The increase ozone tendency due to dynamic in REF is likely associated with an early spring final warming events (**Fig. R4 above**), allowing more waves to break in the stratosphere in late winter and thus, enhances the dynamical ozone transport to the pole during this period. However, since contribution from CHM is dominant during REF, the total net effect is still negative (i.e., less ozone concentration), which is expected during reflective winters. In contrast, during ABS, the final warming is delayed resulting in less dynamical ozone transport to the pole during late winter (**Fig. R4 above**). This is consistent with previous observational studies (e.g., Hu et al., 2014), showing that early spring final warming events that on average occur in early March tend to be preceded by non-SSW winters (i.e., typical of REF winter), while late spring final warming that on average take place up until early May are mostly preceded by SSW events in midwinter (typical of ABS winter). We have modified the text accordingly (**see P13. L25**) and added **Fig. R4** into the supplemental material (**see Fig. S10**).

**4) The positive chemical ozone anomaly in reflective year's midwinter (Figure 12c) is not mentioned or explained in the text.** We have now discussed this in the text (see **P13. L6**).

**5) Please explain if Figure 1 is based on a DWC composite or a single DWC event?**

We have now clarified this. The Fig.1 is based on a DWC composite life cycle. We have also revised the text accordingly.

**6) Line 190: Deceleration doesn't necessarily result in adiabatic cooling.**

We have revised this sentence: "The negative residual circulation anomaly suggests a deceleration of poleward transport of air mass, resulting in negative potential temperature *tendency* over this region" (**see P6, L26**).

**7) Line 211-212: The negative and positive values seemed to be almost at the same level (instead of in the 'upper' and 'lower' stratosphere.)**

Yes, the negative and positive values seemed to be almost at the same level in the stratosphere. We have revised this sentence accordingly (**see P7, L14**).

**8) Line 216: Or transport out of the polar vortex?**
We have revised this sentence (**see P7. L5**).

**9) How were the terms in Figure 2 calculated? Are the DYN and CHM+ANA composites MERRA-2 output? Are the composites in 2d-2f calculated with equation 2? Are the terms consistent (i.e. is the sum of 2d-2f the same as DYN)?**

All the terms are consistent. The dynamical terms in Fig. 2 were calculated using Eq. 2 and Eq. 3. In particular, the sum of the first three right-hand side terms of Eq. 2 is equal to total ozone tendency from dynamics in Eq. 1, while the last term of Eq. 2 is equal to the total ozone tendency due to chemistry and analysis in Eq. 1. We have now clarified this in the text (**see P5. L5**).

**10) Line 395: Is the direct effect mentioned here the effect over the DWC life cycle? This was shown to be nearly zero and not a 'weak increase'.**

We have revised this in the text (**see P13. L5**).

**3.2 Minor comments:**

**1) Line 95: AURA -> Aura**
Corrected.

**2) Line 169: Please explain the minimum ozone term in Table 1. Is this the ozone tendency related to the date of the DWC (and this not necessarily a temporal minimum)?**
This is the minimum value of the total 5-day smoothed ozone tendency from 60-90N during the date of DWC event. We have clarified this in the text (**see P5. L27**).

**3) Line 222: 2e -> 2d**
Corrected.

**4) Caption Figure 1: a-d -> a-b**
Corrected.

---

## Author Comment (AC4) · 12 Dec 2016

Dear Editor and Reviewers

We have seriously considered all minor comments from the three reviewers, and have made substantial changes to the manuscript. As we describe in our detailed responses to the reviewers, we have made numerous modifications that we hope have clarified our paper and improved it as a result.

These changes include:

1. Some new figures in the supplemental material to clarify some reviewer concerns regarding the life cycle of upward wave events (see Figs. S1 and S3) and their relationship with transient ozone changes both in MERRA2 and CESM1 (WACCM) (see Figs. S2 and S4), the life cycle of downward wave-2 events and its relation to transient ozone changes (see Figs. S5-S6), the probability density function distribution and the frequency of extreme heat flux events during REF and ABS (see Fig. S7), evidence for symmetric (linear) response between REF and ABS winters (see Figs. S8-S9), and the time series of the final vortex breakup day in the NH in relation to REF and ABS winters (see Fig. S10).

2. As suggested by reviewer #1 and reviewer #3, we have modified Fig. 1 and Fig. 5 by adding the evolution of wave-1 v'T' and the associated EP flux divergence into this figure. We have now combined the 2 line plots in the old Fig.1 with the line plot in the old Fig. 4, likewise for the model. Finally we have also combined Figs. 9 and 10, and Figs. 11 and 12, for a better comparison between reanalysis and model.

3. We have also modified the schematic figure (Fig. 12) in the manuscript to better highlight our suggested mechanism.

4. A new appendix that highlights the method used for calculating the statistical significance of anomalous values based on a 1000-trial Monte Carlo test is now provided in Appendix C.

We have also made a few changes to the manuscript that is independent of the reviewers' comments. We have uploaded the revised manuscript and new supplementary information along with this response. We hope that all referees find the revised manuscript to be significantly improved and suitable for publication in ACP.

Sincerely,

Sandro Lubis (on behalf of the co-authors)

Please also note the supplement to this comment:
http://www.atmos-chem-phys-discuss.net/acp-2016-558/acp-2016-558-AC4-

supplement.pdf

**Supplement:**

[revised manuscript text omitted]

Manuscript prepared for Atmos. Chem. Phys.
with version 2015/04/24 7.83 Copernicus papers of the LaTeX class copernicus.cls.
Date: 12 December 2016

**Supplementary Material:**

**How Does Downward Planetary Wave Coupling Affect Polar Stratospheric Ozone in the Arctic Winter Stratosphere?**

Sandro W. Lubis[1], Vered Silverman[2], Katja Matthes[1, 3], Nili Harnik[2, 4], Nour-Eddine Omrani[5], and Sebastian Wahl[1]

[1]GEOMAR Helmholtz Centre for Ocean Research Kiel, Germany
[2]Department of Geophysical, Atmospheric and Planetary Sciences, Tel Aviv University, Israel
[3]Christian-Albrechts Universität zu Kiel, Germany
[4]Department of Meteorology, University of Stockholm, Sweden
[5]Geophysical Institute, University of Bergen and Bjerknes Centre for Climate Research, Norway

*Correspondence to:* S. W. Lubis (slubis@geomar.de)

[Figure]

**Figure S1.** Evolution of the upward wave events as a function of time from days -20 to +20 and pressure: (a) wave-1 meridional heat flux anomaly (black contours) and zero contour of the total wave-1 meridional heat flux (blue contour), (b) wave-1 EP flux divergence anomaly, (c) residual mass-streamfunction anomaly, and (d) potential temperature tendency averaged from 60 to 90°N. The black contour intervals are: $\pm$ 1 x $10^9$ [0.5, 1, 2, 4, 8, 16, 32, 64,..] kg s$^{-1}$ for Fig. S1c and $\pm$ 0.5 K day$^{-1}$ for Fig. S1d. The gray shading indicates statistical significance at the 95% level using a 1000-trial Monte Carlo test. The periods of the maximum DWC event (days -3 to +3) are bounded by two vertical red lines.

[Figure]

**Figure S2.** Evolution of the ozone tendencies for the composite upward wave event as a function of time and pressure, averaged from 60 to 90°N: (a) total ozone tendency, (b) ozone tendency anomaly due to dynamics and (c) due to parameterized chemistry. Tendency from the dynamics is decomposed into (d) vertical advection, (e) meridional advection, and (f) eddy transport effects based on Eq.1. The periods of the maximum DWC event (days -3 to +3) are bounded by two vertical red lines. Stippling indicates statistical significance at the 95% level using a 1000-trial Monte Carlo test.

[Figure]

**Figure S3.** As in Fig. S1, but from a 100-yr CESM1(WACCM) simulation. The gray shading indicates statistical significance at the 95% level using a 1000-trial Monte Carlo test. The periods of the maximum DWC event (days -3 to +3) are bounded by two vertical red lines.

[Figure]

**Figure S4.** As in Fig. S2, but from a 100-yr CESM1(WACCM) simulation. The periods of the maximum DWC event (days -3 to +3) are bounded by two vertical red lines. Stippling indicates statistical significance at the 95% level using a 1000-trial Monte Carlo test.

[Figure]

**Figure S5.** Evolution of the downward wave (k=2) events as a function of time from days -20 to +20 and pressure: (a) wave-2 meridional heat flux anomaly (black contours) and (b) wave-2 EP flux divergence anomaly. The black contour intervals are: $\pm\ 1 \times 10^9$ [0.5, 1, 2, 4, 8, 16, 32, 64,..] kg s$^{-1}$ for Fig. S5a and $\pm$ 0.5 K day$^{-1}$ for Fig. S5b. The gray shading indicates statistical significance at the 95% level using a 1000-trial Monte Carlo test.

[Figure]

**Figure S6.** As in Fig. 3, but for wave 2.

[Figure]

**Figure S7.** (a). Probability distribution functions of reflective (blue) and absorptive (red) winters. The vertical black lines represent the 5th (-17.5 Kms$^{-1}$) and 95th (63.6 Kms$^{-1}$) percentile values of the daily distribution from all years. (b) Percentage (frequency) of extreme negative high-latitude averaged wave-1 heat flux events at 50-hPa level vs extreme positive events at the same levels during JFM for all years (black circle), reflective years (blue triangle) and absroptive years (red asterisk).

[Figure]

**Figure S8.** Composite-mean difference (REF or ABS minus climatological mean (CLM)) of the (a,b,g,h) zonal mean ozone, (c,d,i,j) ozone tendency due to dynamics, and (e,f,k,l) ozone tendency due to chemistry from MERRA, averaged between January-February and March-April. Stipplings denote differences significant at the 95% confidence level.

[Figure]

**Figure S9.** As in Fig. S8, but from 100-yr CESM1(WACCM) simulation. Stipplings denote differences significant at the 95% confidence level.

[Figure]

**Figure S10.** (top) Time series of the polar vortex evolution during the time of final vortex breakup in the NH from MERRA2. The final breakup day is defined as the first day when the is negative, and never recovers. (bottom) Composites of the vortex for REF (blue) and ABS (red) winters. Dark green contour lines denote the time series of zonal mean wind from all years.

**Table S1.** As in Table 1, but for the CESM1(WACCM) simulation.

| Dates | $\min_{60-90^\circ N} \overline{v'T'}_{k=1}$ | $\min_{60-90^\circ N} dO_3/dt$ | Dates | $\min_{60-90^\circ N} \overline{v'T'}_{k=1}$ | $\min_{60-90^\circ N} dO_3/dt$ |
|---|---|---|---|---|---|
| 18 Mar 1956 | -41.41 | -23.91 | 19 Feb 2007 | -88.41 | -10.96 |
| 6 Feb 1958 | -61.16 | -25.89 | 28 Feb 2008 | -104.6 | -12.3 |
| 12 Jan 1959 | -54.95 | -7.81 | 16 Jan 2010 | -71.58 | -4.94 |
| 14 Jan 1961 | -125.9 | -5.68 | 5 Mar 2010 | -19.27 | -9.61 |
| 28 Jan 1962 | -92.87 | -32.55 | 3 Feb 2011 | -37.05 | -7.15 |
| 31 Mar 1964 | -68.39 | -3.13 | 26 Mar 2011 | -157.2 | -16.95 |
| 10 Feb 1965 | -144.5 | -47.87 | 5 Jan 2012 | -46.41 | -7.46 |
| 10 Feb 1966 | -64.27 | -9.84 | 29 Jan 2012 | -71.7 | -54.92 |
| 4 Feb 1968 | -59.48 | -4.51 | 1 Jan 2014 | -37.47 | -16.45 |
| 28 Mar 1969 | -54.76 | -16.64 | 16 Feb 2015 | -88.92 | -11.65 |
| 5 Feb 1970 | -91.87 | -16.12 | 25 Jan 2016 | -40.82 | -9.69 |
| 1 Feb 1972 | -38.3 | -4.73 | 17 Mar 2016 | -70.51 | -26.1 |
| 6 Jan 1974 | -44.33 | -8.27 | 8 Jan 2018 | -81.6 | -13.93 |
| 21 Jan 1977 | -63.08 | -5.09 | 15 Feb 2019 | -47.7 | -22.88 |
| 9 Mar 1981 | -45.79 | -17.27 | 23 Mar 2019 | -55.67 | -9.63 |
| 5 Feb 1982 | -69 | -28.15 | 15 Feb 2023 | -70.42 | -31.56 |
| 27 Mar 1982 | -43.4 | -25.63 | 3 Mar 2023 | -82.45 | -56.34 |
| 17 Jan 1984 | -62.91 | -8.39 | 13 Jan 2024 | -38.01 | -13.94 |
| 4 Mar 1984 | -50.41 | -9.26 | 2 Feb 2025 | -103.6 | -6.57 |
| 20 Jan 1985 | -56.24 | -9.08 | 10 Jan 2029 | -87.6 | -16.32 |
| 14 Feb 1985 | -120.2 | -15.66 | 29 Jan 2030 | -101.3 | -8.91 |
| 15 Jan 1986 | -48 | -9.79 | 31 Jan 2031 | -93.74 | -5.11 |
| 2 Feb 1987 | -38.41 | -6.21 | 29 Mar 2034 | -69.24 | -9.68 |
| 10 Mar 1987 | -94.01 | -19.39 | 3 Mar 2037 | -59.01 | -10.83 |
| 26 Feb 1990 | -84.78 | -14.46 | 2 Mar 2038 | -71.45 | -19.67 |
| 19 Jan 1999 | -55.01 | -87.78 | 6 Feb 2042 | -73.8 | -5.08 |
| 11 Feb 1999 | -115.7 | -23.19 | 3 Feb 2049 | -65.93 | -15.85 |
| 3 Jan 2004 | -73.3 | -24.37 | 20 Jan 2054 | -47.82 | -34.82 |
| 11 Jan 2005 | -77.26 | -8.04 | | | |
| 16 Feb 2006 | -90.56 | -47.08 | | | |